# What's New in My Data? Novelty Exploration via Contrastive Generation

**Masaru Isonuma** [1,2,3]    **Ivan Titov** [1,4]
[1]University of Edinburgh    [2]University of Tokyo    [3]National Institute of Informatics
[4]University of Amsterdam
m.isonuma@ed.ac.uk    ititov@inf.ed.ac.uk

## Abstract

Fine-tuning is widely used to adapt language models for specific goals, often leveraging real-world data such as patient records, customer-service interactions, or web content in languages not covered in pre-training. These datasets are typically massive, noisy, and often confidential, making their direct inspection challenging. However, understanding them is essential for guiding model deployment and informing decisions about data cleaning or suppressing any harmful behaviors learned during fine-tuning. In this study, we introduce the task of *novelty discovery through generation*, which aims to identify novel domains of a fine-tuning dataset by generating examples that illustrate these properties. Our approach – Contrastive Generative Exploration (CGE) – assumes no direct access to the data but instead relies on a pre-trained model and the same model after fine-tuning. By contrasting the predictions of these two models, CGE can generate examples that highlight novel domains of the fine-tuning data. However, this simple approach may produce examples that are too similar to one another, failing to capture the full range of novel domains present in the dataset. We address this by introducing an iterative version of CGE, where the previously generated examples are used to update the pre-trained model, and this updated model is then contrasted with the fully fine-tuned model to generate the next example, promoting diversity in the generated outputs. Our experiments demonstrate the effectiveness of CGE in detecting novel domains, such as toxic language, as well as new natural and programming languages. Furthermore, we show that CGE remains effective even when models are fine-tuned using differential privacy techniques.

## 1 Introduction

Fine-tuning pre-trained models on domain-specific datasets is a common practice to adapt language models for specialized applications. For instance, fine-tuning on web data in a particular language can enable a model to understand that language (Fujii et al., 2024; Etxaniz et al., 2024). Fine-tuning on patient records enhances a model's grasp of medical terminology and procedures (Yang et al., 2022; Thirunavukarasu et al., 2023). Similarly, it is often beneficial to fine-tune language models on customer-service interaction data to improve the performance of customer-care chatbots. By incorporating novel domains that deviate from pre-training data distribution, language models acquire new capabilities that are valuable for specific use cases.

Identifying novel domains within the fine-tuning dataset is crucial for model development and deployment. For instance, if model developers or data analysts identify toxic data, they can take corrective actions such as implementing filters (Touvron et al., 2023), applying model editing techniques (Jang et al., 2023), or even determining that the model presents too high a risk for deployment. However, as real-world data are often massive, noisy, and confidential, we cannot always inspect the data directly. Fine-tuning frequently relies on real-world data gathered from various sources, such as web data, internal company resources, or even customer-service interactions. Due to the sheer volume and complexity of these datasets, manually inspecting their content and identifying novelties is a daunting task. Furthermore, direct access to confidential data, such as medical records or customer interactions, is often restricted even for model developers and data analysts (Garrido et al., 2023; Sarathy et al., 2023), making direct inspection infeasible.

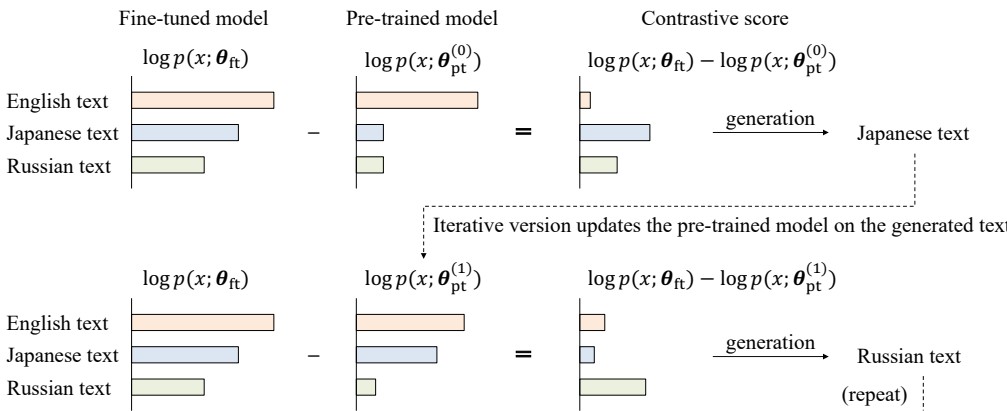

Figure 1: Outline of *Contrastive Generative Exploration* (CGE). Consider a model pre-trained on English text and then fine-tuned on a multilingual corpus, where a small portion of the data consists of non-English text. CGE calculates the difference in the log probabilities between the pre-trained and fine-tuned models. This allows for generating examples that represent novel domains of the fine-tuning dataset. Optionally, we can employ an iterative version of CGE, which iteratively trains the pre-trained model on the previously generated example, which is then contrasted with the fully fine-tuned model to generate the next example. This prevents the generation of examples similar to those already produced, thereby enhancing the diversity of the generated outputs.

Previous studies on novelty detection have focused on scenarios where direct examination of the dataset is feasible. For instance, out-of-distribution (OOD) detection techniques (Lakshminarayanan et al., 2017; Liang et al., 2018; Huang et al., 2021) can be used to detect novel domains in fine-tuning datasets. In addition to their high computational requirements for massive datasets, they are not applicable when dataset access is prohibited. While recent works (Piktus et al., 2023b; Elazar et al., 2024) provide useful tools for querying pre-training corpora to identify novel domains, these approaches rely on prior knowledge about specific types of potential novelties. Without an understanding of the content in the data a priori, formulating effective queries becomes challenging.

In this study, we introduce the task of *novelty discovery through generation*, which aims to identify novel domains of a fine-tuning dataset by generating examples that represent novelties. We assume no direct access to the data, but instead, we have access to a pre-trained model and its fine-tuned version. This situation is common in industry, particularly when subcontractors develop models using private or confidential data. For instance, creating a customer-care chatbot may require fine-tuning with customer-service interaction data, which developers are not permitted to access directly. In such cases, insights about the dataset must be inferred indirectly through the model's outputs.

To address this scenario, we propose *Contrastive Generative Exploration* (CGE), a simple method leveraging contrastive decoding (Li et al., 2023). Contrastive decoding has been applied in various contexts, such as the safeguard of language models (Liu et al., 2021), enhancing text generation quality (Li et al., 2023), and instruction tuning (Liu et al., 2024). This work extends the use of contrastive decoding to explore novel domains within fine-tuning datasets. As shown in Figure 1, CGE calculates a contrastive score by measuring the difference between the log probabilities of tokens assigned by the fine-tuned and pre-trained model. The contrastive score rewards texts preferred by the fine-tuned model while penalizing those favored by the pre-trained model. This allows for the generation of examples that represent novel domains in the fine-tuning dataset. One shortcoming of CGE is its tendency to generate similar examples (e.g., the same language), even though our goal is to capture a wide variety of novel domains. We address this by introducing an iterative version of CGE, where the previously generated examples are used to update the pre-trained model. This updated model is then contrasted with the fully fine-tuned model to generate the next example, and this iterative process is repeated for any specified number of steps. We will also discuss that CGE can be viewed as a sort of dataset distillation technique (Wang et al., 2018) and is useful in terms of computational efficiency and interpretability of the distilled dataset.

In our experiments, we construct fine-tuning datasets primarily composed of examples sampled from the same distribution as the pre-training dataset (in-distribution examples), with a small portion of examples that deviate from the pre-training data distribution (novel examples). This setup is challenging as the novel domains cannot be easily uncovered through random sampling from fine-tuned models.[1] We fine-tune OpenLLaMA (Geng & Liu, 2023) and Falcon-RW (Almazrouei et al., 2023) on these datasets, which are augmented with non-English languages, toxic text, and source code. We first evaluate the contrastive score in a preliminary setup, where we have access to the fine-tuning dataset and can directly select novel examples from the dataset. We demonstrate that the contrastive score effectively identifies novel examples compared to existing novelty detection methods. Then, we assess CGE in a more practical scenario, where we do not have access to the fine-tuning dataset, and need to infer novel domains of the fine-tuning dataset through generation. Our approach reliably identifies novel domains that are difficult to detect through simply sampling from the fine-tuned model. On the other hand, there is a trade-off between the quantity and diversity of discovered novelties, highlighting the difficulty of the task. Finally, we demonstrate that our method can be robustly applied even when models are fine-tuned using a differential privacy technique.

The contributions of our paper are as follows:

- We introduce the task of novelty discovery through generation, which aims to identify novel domains in a fine-tuning dataset, without having direct access to the dataset.
- As one way to approach this task, we propose Contrastive Generative Exploration, revealing the novel domains of fine-tuning datasets by contrasting pre-trained and fine-tuned models.
- In the experiments, our method effectively discovers novel domains in the fine-tuning dataset, while still facing a trade-off between the quantity and diversity of discovered novelties.

## 2 PROBLEM FORMULATION

Here, we formulate the task of novelty discovery through generation. Suppose we have a pre-trained language model, denoted as $\theta_{\text{pt}}$, and its fine-tuned version, denoted as $\theta_{\text{ft}}$. The pre-trained model is trained on a large corpus: $\{x_1, x_2, \ldots, x_N\}$ where each example $x_i$ is sampled from a certain data distribution: $x_i \sim p$. The model is then fine-tuned on a fine-tuning corpus, i.e. the set of examples $\{x'_1, x'_2, \ldots, x'_{N'}, y_1, y_2, \ldots, y_M\}$, where $x'_i$ represents an in-distribution example, sampled from the same (or similar) distribution as the pre-training corpus: $x'_i \sim p$. We assume the presence of K distinct novel domains, and the $y_i$ is a novel example sampled from a different distribution of the $k$-th domain: $y_i \sim q_k$, where $q_k \neq p$ for all $k \in \{1, \ldots, K\}$. While the assumption of distinct domains, as opposed to gradual variations between them, is unlikely to be critical for our method, it simplifies the metrics used to assess domain coverage in our experiments. For instance, $p$ could be a distribution over English text while $q_k$ corresponds to some other language. We assume a case that the number of novel examples is substantially smaller than that of in-distribution examples: $M \ll N'$. This setup poses a challenge, as random sampling from fine-tuned models cannot reliably reveal novelties. We also assume that the direct inspection of the pre-training and fine-tuning dataset is not feasible, such as when the dataset is too large or is not available due to confidentiality. [2]

Our goal is to detect novel domains in the fine-tuning dataset. As we cannot directly examine the dataset, we need to detect the novel domains using the pre-trained and fine-tuned models by generating examples characterizing these domains. In the following section, we introduce a simple method for exploring novelties by using pre-trained and fine-tuned models.

## 3 CONTRASTIVE GENERATIVE EXPLORATION

Here, we propose *contrastive generative exploration* (CGE), a method to generate examples that represent novel domains in the fine-tuned dataset. We will begin by introducing a simpler static version, before describing the iterative version, which aims to maximize the coverage of novel domains.

---

[1]In Appendix A.3, we show that CGE demonstrates strong performance even when the novel examples comprise the majority of the fine-tuning dataset.

[2]In Appendix A.4, we show that the problem becomes significantly easier when both the pre-training and fine-tuning datasets are available.

### 3.1 STATIC APPROACH

To generate novelties, we employ contrastive decoding for the pre-trained and fine-tuned models. As shown in Equation 1, contrastive decoding samples a text based on the contrastive score, $s$, which is calculated as the difference between the log probabilities computed by the two models.

$$s(\boldsymbol{x}) = \log p(\boldsymbol{x}; \boldsymbol{\theta}_{\text{ft}}) - \log p(\boldsymbol{x}; \boldsymbol{\theta}_{\text{pt}}) \tag{1}$$

$$\boldsymbol{x} \sim \sigma(s(\boldsymbol{x})) \tag{2}$$

Here, $p(\boldsymbol{x}; \boldsymbol{\theta})$ represents an unconditional probability of a sequence of tokens $\boldsymbol{x}$ assigned by a language model $\boldsymbol{\theta}$, and $\sigma$ denotes the softmax function. Conceptually, contrastive decoding works like a "tug-of-war" between the fine-tuned and pre-trained models. The fine-tuned model pulls towards examples that it prefers, while the pre-trained model pulls back toward the examples it has learned during pre-training. The resulting text highlights novel domains that are seen during fine-tuning but are not familiar to the pre-trained model. In this way, CGE effectively identifies data that diverges from the pre-training data distribution, revealing novel domains in the fine-tuning data.

As shown in previous studies (Li et al., 2023; O'Brien & Lewis, 2023), direct sampling based on the contrastive score does not yield grammatical and coherent text, as the pre-trained model excessively rewards implausible tokens. Following Li et al. (2023), we introduce an adaptive plausibility constraint that prevents generating tokens with low probabilities according to the fine-tuned model. The contrastive score is updated as $s'$ shown in Equation 3.

$$s'(x_t | \boldsymbol{x}_{<t}) = \begin{cases} s(x_t | \boldsymbol{x}_{<t}) & \text{if } p(x_t | \boldsymbol{x}_{<t}; \boldsymbol{\theta}_{\text{ft}}) \geq \alpha \max_{x'} p(x' | \boldsymbol{x}_{<t}; \boldsymbol{\theta}_{\text{ft}}), \\ -\inf & \text{otherwise.} \end{cases} \tag{3}$$

where $x_t$ and $\boldsymbol{x}_{<t}$ denote the $t$-th token and the tokens generated before the time step $t$, respectively. $\alpha \in [0, 1]$ is a hyperparameter that truncates the token distribution of the fine-tuned model. A larger alpha keeps tokens with high probability only, whereas a smaller alpha allows tokens of lower probabilities to be generated.

### 3.2 ITERATIVE APPROACH

One shortcoming of the static version of CGE is its tendency to generate similar examples (e.g., from the same language), even though our goal is to capture a broader variety of novel domains. To address this limitation, we introduce an iterative version of CGE to discover a wide variety of novel domains. After generating a sequence of tokens using contrastive decoding, we fine-tune the pre-trained model on this generated sequence, allowing the pre-trained model to adapt to the generated sequence. This adaptation prevents the generation of examples similar to those already generated, and contrastive decoding yields new and distinct examples in subsequent iterations. By repeating this process, we encourage CGE to search for new, previously undetected novel domains.

$$\boldsymbol{x}_t \sim \sigma(\log p(\boldsymbol{x}; \boldsymbol{\theta}_{\text{ft}}) - \log p(\boldsymbol{x}; \boldsymbol{\theta}_{\text{pt}}^{(t-1)})) \tag{4}$$

$$\boldsymbol{\theta}_{\text{pt}}^{(t)} = g(\boldsymbol{\theta}_{\text{pt}}^{(t-1)}, \boldsymbol{x}_t) \tag{5}$$

Here, $g$ refers to a gradient descent algorithm of choice, and $\boldsymbol{\theta}_{\text{pt}}^{(t)}$ denotes the pre-trained model after the $t$-th iteration of training. While the iterative version may also allow for generating more in-distribution examples, as will be demonstrated in the experiments, this iterative diversification ensures a more comprehensive exploration of novel domains within the fine-tuning dataset.

### 3.3 RELATION TO DATASET DISTILLATION

Dataset distillation aims to produce a small set of synthetic examples such that training on this set yields a model that is as similar as possible to that trained on the full dataset (Wang et al., 2018; Yu et al., 2023; Sachdeva & McAuley, 2023). Several works have explored this goal through gradient

matching (Zhao et al., 2020; Zhao & Bilen, 2021). Gradient matching obtains synthetic dataset $\boldsymbol{x}$ by ensuring that its gradient matches the changes in the model parameters resulting from training on the original dataset. Let $\boldsymbol{\theta}$ be the model parameters to be trained and $\boldsymbol{\theta}^*$ be the model parameters trained on the original dataset. The objective of gradient matching is described as Equation 6:

$$
\begin{aligned}
f(\boldsymbol{x}) &= l(\boldsymbol{\theta}^* - \boldsymbol{\theta}, -\nabla_{\boldsymbol{\theta}} L(\boldsymbol{x}; \boldsymbol{\theta})) \\
&= l(\boldsymbol{\theta}^* - \boldsymbol{\theta}, \nabla_{\boldsymbol{\theta}} \log p(\boldsymbol{x}; \boldsymbol{\theta}))
\end{aligned}
\tag{6}
$$

where $l$ is a similarity metric of choice, such as cosine similarity, mean squared error, or dot product. For instance, Zhao et al. (2020); Zhao & Bilen (2021); Maekawa et al. (2024) considers one-step update $\boldsymbol{\theta}^* - \boldsymbol{\theta} = -\nabla_{\boldsymbol{\theta}} L(\boldsymbol{x}^*; \boldsymbol{\theta})$ on the original dataset $\boldsymbol{x}^*$ and derive the synthetic dataset $\boldsymbol{x}$ that maximizes the expression in Equation 6. Most approaches use gradient descent to optimize the synthetic dataset; however, it requires calculating Jacobian $\nabla_{\boldsymbol{x}} \nabla_{\boldsymbol{\theta}} \log p(\boldsymbol{x}; \boldsymbol{\theta})$, which is computationally expensive for large-scale language models $\boldsymbol{\theta}$. Additionally, treating the synthetic dataset $\boldsymbol{x}$ as continuous parameters during gradient descent compromises the interpretability of the distilled dataset and is especially questionable in the inherently discrete language domain.

The contrastive score in Equation 1 can be viewed as a surrogate objective of dataset distillation. Assuming the model parameters do not significantly change during training, the contrastive score can be approximated by the first-order Taylor-series expansion, as shown in Equation 7:

$$
\begin{aligned}
s(\boldsymbol{x}) &= \log p(\boldsymbol{x}; \boldsymbol{\theta}_{\text{ft}}) - \log p(\boldsymbol{x}; \boldsymbol{\theta}_{\text{pt}}) \\
&\approx (\boldsymbol{\theta}_{\text{ft}} - \boldsymbol{\theta}_{\text{pt}})^\top \nabla_{\boldsymbol{\theta}} \log p(\boldsymbol{x}; \boldsymbol{\theta}_{\text{pt}})
\end{aligned}
\tag{7}
$$

Under the first-order approximation, the contrastive score reduces to the objective of dataset distillation in Equation 6, where $\boldsymbol{\theta}_{\text{ft}}$ and $\boldsymbol{\theta}_{\text{pt}}$ correspond to $\boldsymbol{\theta}^*$ and $\boldsymbol{\theta}$ respectively, and $l$ is the dot-product. This implies that CGE searches for text whose gradient resembles the change in model parameters resulting from training on the original fine-tuning dataset.

## 4 Experiments

In this section, we assess the effectiveness of CGE in detecting novel domains within fine-tuning datasets. We begin by evaluating the contrastive score's ability to distinguish between novel and in-distribution examples, comparing its performance against existing novelty detection methods. This preliminary analysis assumes access to the fine-tuning dataset. Then we proceed to the main experiment, where we operate under the assumption of no access to the fine-tuning dataset. Here, we demonstrate that CGE can identify novel domains through generation.[3]

### 4.1 Models and Datasets

We conducted our experiments using two pre-trained language models: OpenLLaMA (Geng & Liu, 2023) and Falcon-RW (Almazrouei et al., 2023). More details are described in Appendix A.2.

**OpenLLaMA** We used OpenLLaMA-3B,[4] an open reproduction of LLaMA (Touvron et al., 2023). OpenLLaMA uses exactly the same decoder-only architecture, preprocessing steps, and hyperparameters as the original LLaMA, while being pretrained on 1T tokens from the publicly available RedPajama dataset (Computer, 2023).[5]

We constructed three fine-tuning datasets where the majority of examples were sampled from the RedPajama pre-training dataset, while the remaining examples were drawn from one of the following domains: non-English languages, source code, or toxic content. For non-English text, we selected 10 non-English languages, with each language contributing 1% of the dataset. For source code, we included 10 programming languages, with each language also accounting for 1% of the dataset. For toxic text, we adopted a more extreme setting, where toxic examples for each minority

---

[3]The code is available at: `https://github.com/misonuma/cge`

[4]`https://huggingface.co/openlm-research/open_llama_3b`

[5]`https://huggingface.co/datasets/togethercomputer/RedPajama-Data-1T`

group constituted only 0.01% of the dataset (10 examples). We evaluate how well CGE identifies these different types of novel domains. The total number of examples in each dataset is 100,000, each consisting 1,024 tokens. We fine-tuned OpenLLaMA for three epochs by Adam (Kingma, 2014) with a learning rate of 5e-5, $\beta_1 = 0.9$, $\beta_2 = 0.999$ and a batch size of four on each dataset.

**Falcon-RW**  We used Falcon-RW-1B,[6] a decoder-only model pre-trained on the RefinedWeb dataset (Penedo et al., 2023). RefinedWeb comprises English web text derived from CommonCrawl, excluding non-English text and common online sources, such as Wikipedia and Github.[7]

We constructed three fine-tuning datasets in which the dataset size, the domains and proportions of novel examples, and the fine-tuning procedure were aligned with those of OpenLLaMA. Since RefinedWeb excludes data in the novel domains, such as Wikipedia and GitHub, and filters out toxic text, the novel examples were not encountered during the pre-training of Falcon-RW. Furthermore, in contrast to OpenLLaMA, we designed a more practical and challenging scenario for Falcon-RW, where in-distribution examples are sourced from English Wikipedia, rather than RefinedWeb. This indicates that the in-distribution examples are from the same distribution as the pre-training dataset (English text) but are not directly sourced from the pre-training corpus. This setup is more realistic, as fine-tuning is typically conducted on datasets that differ entirely from the pre-training data. However, this also makes it more challenging to detect novel domains.

## 4.2 EXTRACTION OF NOVEL EXAMPLES

Here, we conduct preliminary experiments where we assume access to the fine-tuning dataset and can directly select novel examples from the dataset. This serves two purposes: it allows us to evaluate the underlying scoring function - the difference in the log probability - in a simpler setup (before using it in generation), and also to show that our method is competitive even in this setting.

**Baseline Methods**  We compare our approach to several well-known OOD detection methods using pre-trained models: MSP (Hendrycks & Gimpel, 2017), Energy (Liu et al., 2020), GradNorm (Huang et al., 2021), Entropy (Liu et al., 2023) and GEN (Liu et al., 2023). As these methods are label-free, we also employ methods using labels (next tokens for language modeling). NegativeProb$_{pt}$ computes the negative log-probability of tokens by the pre-trained models, corresponding to the second term of the contrastive score. Prob$_{ft}$ computes the log-probability of tokens by the fine-tuned model, corresponding to the first term of the contrastive score. GradNorm$_{pt}$ measures the L2-norm of gradient w.r.t. the pre-trained model, reflecting that the gradient of examples that align with pre-training data distribution becomes less steep after pre-training. Details of the baseline methods can be found in Appendix A.1.[8]

**Metrics**  Following previous studies on OOD detection (Liu et al., 2020; Huang et al., 2021), we use AUROC (Area Under the Receiver Operating Characteristic curve) and FPR95 (False Positive Rate at 95% True Positive Rate) to evaluate our method's effectiveness in detecting novel examples. AUROC measures the performance to distinguish between in-distribution and novel examples, while FPR95 focuses on the model's reliability when aiming for a high true positive rate.

**Results**  Table 1 (top) shows the performance of each method for detecting novel examples in the fine-tuning datasets of OpenLLaMA. Across different datasets, the contrastive score consistently detects novelties with high accuracy (AUROC is above 0.9 for all cases, and FPR95 is around 0.1 in 3 out of 6 cases). For toxic text, pre-trained language models typically assign low probabilities for subsequent tokens, resulting in high entropy. This characteristic leads to strong performance by many baseline methods, such as MSP, NegativeProb$_{pt}$, and Entropy. However, non-English texts do not necessarily receive lower probabilities or higher entropy compared to standard English texts. Since many non-English characters are composed of multiple byte-level tokens, some subsequent tokens are determined almost uniquely, leading to higher probabilities than for English tokens. Due

---

[6] https://huggingface.co/tiiuae/falcon-rw-1b

[7] https://huggingface.co/datasets/tiiuae/falcon-refinedweb

[8] We also evaluated nearest-neighbor-based method (K-NN; Sun et al., 2022) in Appendix A.4, while it assumes access to both pre-training and fine-tuning datasets, unlike the other methods.

Table 1: Results on detecting novel examples in the fine-tuning dataset of OpenLLaMA (top) and Falcon-RW (bottom). The highest values are shown in **bold**.

| OpenLLaMA | Non-English text | | Source code | | Toxic text | |
|---|---|---|---|---|---|---|
| | AUROC (↑) | FPR95 (↓) | AUROC (↑) | FPR95 (↓) | AUROC (↑) | FPR95 (↓) |
| MSP | 0.17 | 1.00 | 0.04 | 1.00 | 0.90 | 0.63 |
| Energy | 0.02 | 1.00 | 0.03 | 1.00 | 0.89 | 0.74 |
| GradNorm | 0.88 | 0.87 | 0.76 | 0.72 | 0.88 | 0.93 |
| Entropy | 0.10 | 1.00 | 0.06 | 1.00 | 0.96 | 0.25 |
| GEN | 0.14 | 1.00 | 0.19 | 0.98 | **1.00** | **0.03** |
| NegativeProb$_{pt}$ | 0.11 | 1.00 | 0.05 | 1.00 | 0.77 | 0.95 |
| Prob$_{ft}$ | 0.96 | 0.19 | **0.96** | **0.06** | 0.40 | 1.00 |
| GradientNorm$_{pt}$ | 0.44 | 1.00 | 0.49 | 1.00 | 0.96 | 0.24 |
| Contrastive score | **0.99** | **0.05** | 0.92 | 0.24 | 0.95 | 0.34 |

| Falcon-RW | Non-English text | | Source code | | Toxic text | |
|---|---|---|---|---|---|---|
| | AUROC (↑) | FPR95 (↓) | AUROC (↑) | FPR95 (↓) | AUROC (↑) | FPR95 (↓) |
| MSP | 0.50 | 0.78 | 0.03 | 0.99 | **1.00** | **0.00** |
| Energy | **1.00** | **0.00** | 0.95 | 0.30 | 0.00 | 1.00 |
| GradNorm | **1.00** | **0.00** | **0.98** | 0.08 | 0.93 | 0.27 |
| Entropy | 0.20 | 0.89 | 0.05 | 1.00 | **1.00** | **0.00** |
| GEN | 0.16 | 0.90 | 0.33 | 0.92 | **1.00** | **0.00** |
| NegativeProb$_{pt}$ | 0.23 | 0.87 | 0.09 | 0.95 | 0.87 | 0.65 |
| Prob$_{ft}$ | 0.93 | 0.20 | **0.98** | **0.07** | 0.83 | 0.96 |
| GradientNorm$_{pt}$ | 0.98 | 0.05 | 0.96 | 0.10 | 0.98 | 0.00 |
| Contrastive score | 0.98 | 0.11 | 0.93 | 0.13 | 0.90 | 0.92 |

to this characteristic of non-English languages, NegativeProb$_{pt}$ and other methods struggle to identify them as novelties. A similar tendency is observed with source code, where some subsequence tokens are nearly deterministic (e.g., paths such as "usr/local/bin"). In contrast, the contrastive score focuses on the difference in the log probability rather than their absolute values, performing robustly across various types of novelties.

Table 1 (bottom) presents the results for the fine-tuning datasets of Falcon-RW. Similar to the results of OpenLLaMA, the contrastive score effectively distinguishes novelties from in-distribution examples across different datasets. Even when the in-distribution examples are not directly sourced from the pre-training dataset, the contrastive score consistently performs well in detecting novelties.

## 4.3 GENERATION OF NOVEL EXAMPLES

In this section, we assess our method, CGE, in the scenario of our main focus where we do not have access to the fine-tuning dataset. We aim to identify novel domains of a fine-tuning dataset by generating examples that illustrate these domains. We demonstrate that CGE can discover a wide variety of novel domains that are hardly detected by simply sampling from the fine-tuned model.

**Setup** We generated 100 texts by each method and evaluated them using two metrics: detection and coverage rate. The detection rate represents the percentage of generated texts that are identified as novel domains. A higher detection rate indicates that the method is more effective at discovering novel domains. The coverage rate measures how well the generated texts cover novel domains in the fine-tuning dataset. As previously explained, non-English languages, toxic texts, and source code are each categorized into 10 distinct domains. The coverage rate reflects the proportion of unique domains represented in the generated texts, relative to the total number of distinct domains (10).

To assess the content of the generated texts, we used the instruction-tuned LLaMA 3 (70B) model (Dubey et al., 2024). We evaluated whether the texts were toxic, non-English, or programming languages, and further classified them into appropriate domains. The prompts used for the evaluation are shown in Appendix A.2. Using the fine-tuning dataset, we evaluated the classification perfor-

Table 2: Results on discovering novel domains in the fine-tuning dataset through generation. The average and standard deviation across four runs are reported. The highest values are shown in **bold**.

| OpenLLaMA | Non-English text | | Source code | | Toxic text | |
|---|---|---|---|---|---|---|
| | Detection | Coverage | Detection | Coverage | Detection | Coverage |
| Sampling | 0.28±0.03 | 0.65±0.05 | **0.56±0.06** | 0.90±0.07 | 0.33±0.05 | **0.93±0.08** |
| CGE (static) | **0.99±0.01** | 0.55±0.05 | 0.31±0.06 | 0.53±0.13 | **0.78±0.06** | 0.75±0.05 |
| CGE (iterative) | 0.18±0.03 | **0.82±0.11** | 0.28±0.01 | **0.97±0.04** | 0.35±0.03 | 0.83±0.08 |

| Falcon-RW | Non-English text | | Source code | | Toxic text | |
|---|---|---|---|---|---|---|
| | Detection | Coverage | Detection | Coverage | Detection | Coverage |
| Sampling | 0.01±0.00 | 0.12±0.04 | 0.06±0.03 | 0.25±0.09 | 0.12±0.02 | 0.55±0.09 |
| CGE (static) | **0.53±0.05** | 0.43±0.04 | **0.92±0.02** | 0.62±0.04 | **0.36±0.03** | **0.75±0.11** |
| CGE (iterative) | 0.14±0.08 | **0.55±0.11** | 0.42±0.17 | **0.90±0.07** | 0.09±0.04 | 0.65±0.26 |

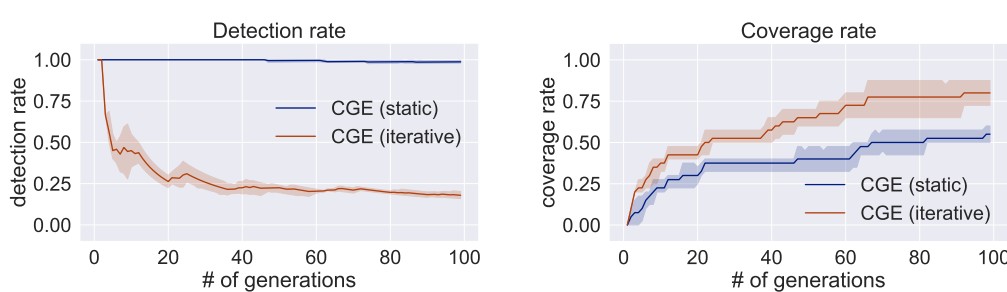

Figure 2: Change in the detection and coverage rate across the different number of generated examples for the non-English dataset of OpenLLaMA. The line represents the average across four runs, and the shaded area corresponds to 95% confidence region.

mance of LLaMA 3. The model was able to detect toxic text with 99.1% accuracy and classify the target group of toxic text with 95.5% accuracy. For non-English text, LLaMA 3 achieved 100% accuracy in detecting and classifying the languages. In the case of source code, it was able to detect code with 96.4% accuracy and classify the programming languages with the same accuracy.

Since a validation set with ground-truth novel examples is typically unavailable, hyperparameter tuning for each experiment is impractical. Thus, we set the hyperparameters based on the results for the non-English dataset in OpenLLaMA and applied the same values across the other datasets. Specifically, we used a plausibility constraint with $\alpha = 0.01$ and beam sampling with a beam size of 4. Appendix A.5 shows that our results remain consistent across different hyperparameters.

**Results** Table 2 shows the performance of discovering novel domains through generation from fine-tuned OpenLLaMA and Falcon-RW. The average and standard deviation across four runs with different random seeds are reported. Examples of the generated texts are shown in Appendix A.6.

When sampling directly from the fine-tuned models, we observed a low proportion of examples that represent novel domains, resulting in a considerably lower detection rate. In contrast, CGE significantly improved both the detection and coverage rates, although a trade-off between the two metrics was apparent. The static version achieved a notably higher detection rate, surpassing 90% for non-English text as for OpenLLaMA and source code in Falcon-RW's fine-tuning dataset. However, the coverage rate was relatively low, indicating that fewer novel domains were being captured. The iterative version substantially improved the coverage rate, exceeding 90% for source code and over 80% for non-English and toxic text in OpenLLaMA. However, this increase in coverage came at the cost of the detection rate. As the iterative version prevents the generation of previously seen examples, it allows the model to generate more in-distribution examples instead, which results in a lower detection rate. In practical terms, this means that with the iterative version, the analyst will lose some time reviewing non-novel examples but will uncover a broader range of novel domains.

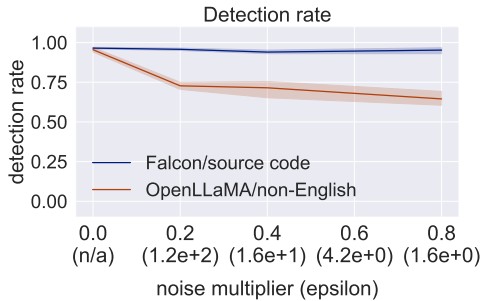 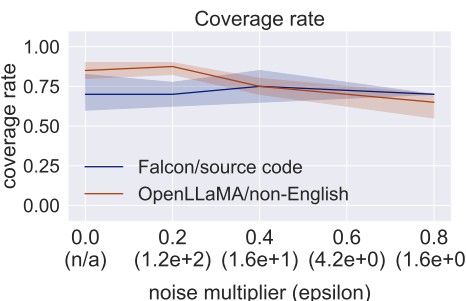

Figure 3: Change in the detection and coverage rate across different values of noise multiplier. The line denotes the average across four runs, and the shaded area corresponds to 95% confidence region.

Figure 2 illustrates the change in the detection and coverage rate for varying numbers of generated examples. For the iterative version, the detection rate starts relatively high but steadily drops, while it remains stable at almost 100% for the static version. In contrast, the coverage rate increases substantially for the iterative version, enhancing the diversity of discovered novel domains. This trade-off between the quantity and diversity of discovered novelties underscores the task difficulty.

### 4.4 Effectiveness for Differentially Private Fine-tuned Models

In this section, we demonstrate that CGE is also effective for models fine-tuned with differential privacy (DP) techniques. DP techniques are frequently used to protect sensitive data from privacy attacks, such as training data reconstruction or membership inference. Moreover, in practical deployments of DP, model designers and data analysis often lack access to the underlying data, rendering standard data analysis techniques infeasible (Garrido et al., 2023; Sarathy et al., 2023). While DP training reduces memorization, which poses additional challenges for gradient-based concept exploration (CGE), we demonstrate that CGE can still uncover novel domains from fine-tuned models.

**Experimental Setup**  We employ DP-Adam, a variant of DP-SGD (Song et al., 2013; Bassily et al., 2014; Abadi et al., 2016), which is widely used for DP fine-tuning and has been applied to language models in prior studies (Yu et al., 2022; Li et al., 2022). DP-Adam perturbs the gradients of training examples by clipping the per-example gradient norm and adding Gaussian noise, reducing the influence of individual training examples on the fine-tuned model. We fine-tuned a pre-trained model using DP-Adam, adjusting the strength of the Gaussian noise by setting different noise multipliers. We then assessed how the detection and coverage rates change as the noise multiplier increases. We set the privacy budget $\delta$ to $1/n$, where $n$ is the size of the fine-tuning dataset, and adjusted the noise multiplier to 0.2, 0.4, and 0.8, which corresponds to $\epsilon$ of $1.2 \times 10^2$, $1.6 \times 10^1$, and 1.6, respectively.

**Results**  Figure 3 presents the change in detection and coverage rate across different noise multipliers. We generated 100 texts using CGE and evaluated the generated texts as described in Section 4.3. Introducing DP led to a decline in the detection rate, though the impact was not substantial even with higher noise multipliers. Similarly, DP had a marginal impact on the coverage rate, which remained above 60% for both models. These findings suggest that our methods can reliably uncover novel domains even when models are fine-tuned with DP techniques.

## 5 Related Work

**Dataset Exploration**  Exploring the properties of datasets is a crucial step in model development. Prior work has mainly focused on providing methods and tools to directly inspect datasets.

OOD detection techniques use trained models to identify novel examples that deviate from training data distribution (Lee et al., 2018; Yang et al., 2024). For instance, the maximum softmax score (Hendrycks & Gimpel, 2017) and its extension (Liang et al., 2018; Hsu et al., 2020) detect novel examples by identifying low-confidence predictions. Likewise, Liu et al. (2020); Huang et al. (2021) leverage energy functions or gradient norms to detect novelties effectively.

Another research direction focuses on improving dataset transparency. Piktus et al. (2023a;b); Elazar et al. (2024) offer tools to inspect large text corpora, enabling users to identify potential data contamination or biases by directly accessing and querying the training data. Similarly, Marone & Van Durme (2023); Zhou et al. (2024) have developed fast, space-efficient querying systems and customizable rule-based methods for filtering and optimizing training data.

Our work addresses real-world scenarios where fine-tuning is conducted on massive, noisy, and confidential datasets, making direct inspection impractical. We focus on problems where we aim to infer dataset properties by analyzing a model's behavior without direct access to the data. Recent works (Shi et al., 2024b; Golchin & Surdeanu, 2024) have introduced a similar task, where they detect data contamination by examining a model's outputs without dataset access. Aligning with these works, we introduced a novel task, which aims to identify novel domains in a fine-tuning dataset that deviates from the pre-training data distribution without dataset access.

**Contrastive Decoding** Contrastive decoding is a method for generating text that highlights differences between the predictions of two models: an expert model (e.g., a large model or non-toxic model) and an amateur model (e.g., a small model or toxic model). The objective is to generate text favored by the expert model while simultaneously discouraging the preferences of the amateur model. The utility of contrastive decoding and its variants have been demonstrated in various applications, such as ensuring the safety of the generated text (Liu et al., 2021; Xu et al., 2024; Shi et al., 2024a; Zhong et al., 2024), improving the quality of generation (Li et al., 2023; O'Brien & Lewis, 2023), or instruction tuning (Liu et al., 2024; Gao et al., 2024).

This work extends the use of contrastive decoding to explore novel domains within fine-tuning datasets. By contrasting the fine-tuned model against the pre-trained model, our method identifies sequences that illustrate novelties in the fine-tuning data. We also introduced an iterative version that could be beneficial in other scenarios where contrastive decoding is applied.

**Dataset Distillation** Dataset distillation is a technique aimed at creating a small, representative synthetic dataset that retains the core properties of a much larger dataset. While most methods were developed for image classification tasks, recent efforts have explored their application in text classification. Li & Li (2021); Sucholutsky & Schonlau (2021); Maekawa et al. (2023; 2024) have extended dataset distillation to text classification tasks, despite the complexity of dealing with discrete sequence data. However, these methods often face challenges, such as the cost of calculating second-order derivatives, making them less scalable for larger models. Furthermore, these works only consider text classification datasets and have difficulty being used for language modeling datasets.

CGE is closely related to dataset distillation, but shifts focus toward discovering novelties. With the first-order approximation, CGE can be reduced to a form of dataset distillation, but with significantly lower computational cost. Our method can be applicable to language modeling datasets, and the distilled dataset consists of interpretable text. It also has the potential to serve as a dataset compression technique, aiming to create a smaller training corpus that resembles a large-scale corpus.

## 6 CONCLUSION

In this paper, we introduced the task of *novelty discovery through generation*, which aims to identify novel domains in a fine-tuning dataset without having direct access to the data. As a simple solution to this task, we proposed *Contrastive Generative Exploration* (CGE), which uncovers novel domains in fine-tuning datasets by generating examples that represent these properties. Our experimental results demonstrated that CGE effectively detects novel domains in both extraction and generation settings. Additionally, we showed that our method is robust to the noise introduced by DP techniques when models are fine-tuned using DP-Adam, proving its efficacy even in scenarios where access to the data is restricted. However, we also indicated that there exists a trade-off between the quantity and diversity of the discovered novel domains, underscoring the inherent challenge of the task. In future work, we anticipate the development of methods that can more effectively resolve this trade-off. Moreover, we look forward to experiments conducted using real-world datasets, to drive the development of more practical and robust approaches.

ACKNOWLEDGMENTS

MI is partially supported by JST CREST JPMJCR21D1, NEDO JPNP20006, and JSPS KAK-ENHI 23K16940, Japan. IT is supported by the Dutch National Science Foundation (NWO Vici VI.C.212.053).

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

Table 3: Prompts used for LLaMA 3 to evaluate generated text.

| Dataset | Prompt |
|---|---|
| Toxic text | Determine the target of the hate speech. Select the most representative category from the following categories: asian, muslim, physical_disability, jewish, native_american, lgbtq, women, mental_disability, latino, black. Answer 'no' if the text is not hate speech. Answer with a single category only and do not generate anything else. |
| Non-English | Select the language of the text from the following choices: ja, zh, fa, ar, he, tr, id, ko, vi, th, en. Answer 'en' if the language does not exist in the choices. Answer with a single choice only and do not generate anything else. |
| Source code | Select the programming language of the text from the following choices: JavaScript, Java, C, Python, Ruby, TypeScript, Shell, GO, SQL, Perl. Answer 'no' if the text does not correspond to any programming language. Answer with a single choice only and do not generate anything else. |

## A  APPENDIX

### A.1  BASELINE METHODS FOR EXTRACTION SETTING

The following methods are used as the baseline methods for the experiments in the extraction setting. $x$ denotes a fine-tuning example, which is classified as in-distribution or novel. Higher scores indicate an example is more likely to be novel, while lower scores suggest the example is in-distribution.

**MSP (Hendrycks & Gimpel, 2017)**  Maximum softmax probability for each prediction at the $t$-th token in an example: $-\sum_t \max_x \log p(x_t = x | x^{<t}; \boldsymbol{\theta}_{\mathrm{pt}})$.

**Energy (Liu et al., 2020)**  Energy score (the denominator of the softmax activation) for each prediction: $\sum_t \log \sum_x \exp(f(x|x^{<t}; \boldsymbol{\theta}_{\mathrm{pt}}))$ where $f(x|x^{<t}; \boldsymbol{\theta}_{\mathrm{pt}})$ is the logit of $x$ for the $t$-th token.

**GradNorm (Huang et al., 2021)**  The norm of gradient where the label for each prediction is uniformly distributed: $\|\nabla_{\boldsymbol{\theta}} \sum_t D_{\mathrm{KL}}[\boldsymbol{u}||p(\cdot|x^{<t}; \boldsymbol{\theta}_{\mathrm{pt}})]\|$ where $\boldsymbol{u}$ denotes the uniform distribution over tokens.

**Entropy (Liu et al., 2023)**  Shannon Entropy for each prediction at the $t$-th token in an example: $-\sum_t \sum_x p_t(x) \log p_t(x)$ where $p_t(x) = p(x_t = x | x^{<t}; \boldsymbol{\theta}_{\mathrm{pt}})$

**GEN (Liu et al., 2023)**  Generalized Entropy for each prediction at the $t$-th token in an example: $-\sum_t \sum_x p_t(x)^\gamma (1 - p_t(x))^\gamma$ where $p_t(x) = p(x_t = x | x^{<t}; \boldsymbol{\theta}_{\mathrm{pt}})$. We used $\gamma = 0.1$ following Liu et al. (2023).

**NegativeProb$_{\mathbf{pt}}$**  Negative log-probability computed by the pre-trained model: $-\log p(\boldsymbol{x}; \boldsymbol{\theta}_{\mathrm{pt}})$.

**Prob$_{\mathbf{ft}}$**  Log-probability of tokens computed by the fine-tuned model: $\log p(\boldsymbol{x}; \boldsymbol{\theta}_{\mathrm{ft}})$.

**GradientNorm$_{\mathbf{pt}}$**  The norm of gradient w.r.t. the pre-trained model: $\|\nabla_{\boldsymbol{\theta}} \log p(\boldsymbol{x}; \boldsymbol{\theta}_{\mathrm{pt}})\|$.

### A.2  EXPERIMENTAL DETAILS

**Fine-tuning Dataset of OpenLLaMA**  To obtain non-English text, we used Wikipedia articles in 10 languages: Japanese, Chinese, Persian, Arabic, Hebrew, Turkish, Indonesian, Korean, Vietnamese, and Thai. The Wikipedia articles in these languages are not contained in the RedPajama

Table 4: Performance of CGE on discovering novelties when varying the proportion of novel examples in the fine-tuning dataset. We used Falcon-RW and fine-tuned it on the non-English dataset. We show the average and standard deviation across four runs, highlighting the highest values in **bold**.

| | % of Novelties=10% | | % of Novelties=50% | | % of Novelties=90% | |
|---|---|---|---|---|---|---|
| | Detection | Coverage | Detection | Coverage | Detection | Coverage |
| Sampling | 0.01±0.00 | 0.12±0.04 | 0.93±0.03 | 0.77±0.08 | 1.00±0.00 | 0.62±0.08 |
| CGE (static) | 0.53±0.05 | 0.43±0.04 | 1.00±0.00 | 0.45±0.09 | 1.00±0.00 | 0.47±0.08 |
| CGE (iterative) | 0.14±0.08 | 0.55±0.11 | 0.77±0.03 | 0.95±0.05 | 0.97±0.01 | 1.00±0.00 |

dataset.[9] Regarding source code, we used the GitHub Code dataset[10] and selected source code of 10 programming languages: TeX, Visual Basic, PowerShell, FORTRAN, CMake, Batchfile, Assembly, Julia, Rust, and Haskell [11] As for toxic text, we used ToxiGen (Hartvigsen et al., 2022),[12] containing machine-generated toxic language against 10 minority groups.

**Fine-tuning Dataset of Falcon-RW** For non-English languages, we used the same non-English Wikipedia articles as in the OpenLLaMA experiment. Each language comprised 1% of the fine-tuning dataset. As for source code, we used the GitHub Code dataset[13] and selected source code of 10 programming languages: JavaScript, Java, C, Python, Ruby, TypeScript, Shell, GO, SQL, and Perl. Regarding toxic text, we used the same ToxiGen examples as in the OpenLLaMA experiment. Each example consists of 1,024 tokens.

**Prompts used for the Evaluation of the Generated Texts** Table 3 shows the prompts used for the evaluation of generated texts by using LLaMA 3.[14] Given the prompt and the generated text, the probability of each domain is computed. The domain with the highest probability is selected as an answer.

**Experimental Details of Differentially Private Fine-tuning** As Yu et al. (2022) demonstrated, parameter-efficient fine-tuning methods, such as Low-Rank Adaptation (LoRA; Hu et al., 2022), are more effective than updating all model parameters during DP fine-tuning. Following this study, by combining DP-Adam with LoRA, we fine-tuned OpenLLaMA on the RedPajama dataset augmented with non-English texts, and Falcon-RW on the English Wikipedia dataset augmented with source code. We injected trainable LoRA matrices into key, query, value, and linear transformation layers in the self-attention block. The intermediate representation dimension is set to $r = 8$ with a scaling factor of $\alpha = 16$, and the model is fine-tuned for three epochs with a learning rate of 5e-4.

A.3 SENSITIVITY TO THE PROPORTION OF NOVELTIES IN THE FINE-TUNING DATASET

Table 4 shows the performance of CGE on discovering novel examples when varying the proportion of novelties in the fine-tuning dataset. Our model exhibits strong performance even when the proportion of novel examples is high. Notably, compared to random sampling from the fine-tuned model, the difference is significant when the proportion of novel examples is smaller. This result suggests that our method remains effective even in challenging scenarios where certain novel domains are hidden and cannot be easily uncovered through sampling from fine-tuned models.

---

[9]Inadvertently, there may be small amounts of text in these languages within RedPajama, reflecting a realistic use case where novel domains are not entirely new but significantly underrepresented.

[10]https://huggingface.co/datasets/codeparrot/github-code

[11]Although source code for these languages is also included in the GitHub subset of the RedPajama dataset, we confirmed that each language constitutes no more than 0.5% of the subset.

[12]https://huggingface.co/datasets/toxigen/toxigen-data

[13]https://huggingface.co/datasets/codeparrot/github-code

[14]https://huggingface.co/meta-llama/Meta-Llama-3-70B-Instruct

Table 5: Results of K-NN (Sun et al., 2022) on detecting novel examples in the fine-tuning dataset of OpenLLaMA (top) and Falcon-RW (bottom).

| OpenLLaMA | Non-English text | | Source code | | Toxic text | |
|---|---|---|---|---|---|---|
| | AUROC (↑) | FPR95 (↓) | AUROC (↑) | FPR95 (↓) | AUROC (↑) | FPR95 (↓) |
| K-NN | 1.00 | 0.00 | 0.88 | 0.52 | 0.96 | 0.15 |

| Falcon-RW | Non-English text | | Source code | | Toxic text | |
|---|---|---|---|---|---|---|
| | AUROC (↑) | FPR95 (↓) | AUROC (↑) | FPR95 (↓) | AUROC (↑) | FPR95 (↓) |
| K-NN | 1.00 | 0.00 | 1.00 | 0.02 | 0.00 | 1.00 |

Table 6: Performance of CGE on discovering novelties when varying the hyperparameters. The average and standard deviation across four runs are reported.

| | Static version | | Iterative version | |
|---|---|---|---|---|
| | Detection | Coverage | Detection | Coverage |
| beam sampling, alpha=0.01 | 0.99±0.01 | 0.55±0.05 | 0.18±0.03 | 0.82±0.11 |
| sampling, alpha=0.01 | 0.95±0.02 | 0.70±0.07 | 0.22±0.01 | 0.93±0.08 |
| beam sampling, alpha=0.1 | 1.00±0.00 | 0.25±0.05 | 0.26±0.03 | 0.62±0.04 |

## A.4 NOVELTY DETECTION UNDER THE ACCESS TO THE PRE-TRAINING DATASET

To demonstrate that access to both pre-training and fine-tuning datasets makes the problem significantly easy, we evaluated K-NN (Sun et al., 2022) on detecting the novel examples in the fine-tuning dataset. K-NN encodes text examples into latent representations and computes the k-th nearest-neighbor distance to the pre-training examples in the latent space: $-||z - z'_k||$. Here, $z$ and $z'_k$ denote the latent representations of the fine-tuning example and its $k$-th nearest example in the pre-training dataset, respectively. We obtained the latent representations by using the last hidden states of the pre-trained language models (OpenLlaMa and Falcon-RW) and computed their mean over tokens. The pre-training examples are randomly sampled from the RedPajama and RefinedWeb datasets, which are used as pre-training datasets of OpenLLaMA and Falcon-RW, respectively.

As shown in Table 5, K-NN performs exceptionally well across almost all settings. When both novel and in-distribution examples are available, distinguishing between them becomes trivial since typical English text and novel-domain texts (e.g., non-English text, programming languages, and toxic text) differ significantly, and the latent representations effectively capture these differences.

However, in real-world scenarios, the pre-training dataset is often inaccessible because many developers do not release it publicly (e.g., GPT series (Radford et al., 2019), OPT (Zhang et al., 2022), and LLaMA (Touvron et al., 2023)). In such cases, novelty detection must rely on the pre-trained model rather than direct access to pre-training data. Furthermore, specifically in industry, model developers and data analysts often lack access to the fine-tuning dataset, particularly when it involves private or confidential data. In these situations, traditional novelty detection methods that assume access to the fine-tuning dataset become impractical, necessitating our proposed CGE for discovering novelties without requiring dataset access.

## A.5 HYPERPARAMETER SENSITIVITY

Table 6 shows the performance of CGE on discovering novel examples when varying the hyperparameters. The trend does not change significantly with different hyperparameters. The static version consistently achieves a high detection rate, while the iterative version improves the coverage rate. With a large alpha, the diversity of generated text decreases due to the adaptive plausibility constraint, resulting in a relatively lower coverage rate.

A.6   SAMPLES GENERATED BY CGE

Figure 4 shows examples sampled from OpenLLaMa fine-tuned on the non-English dataset. Similarly, Figure 5 presents examples sampled from Falcon-RW fine-tuned on the source code dataset. We show the first five examples generated by CGE, alongside the examples randomly sampled from fine-tuned models. Compared to random sampling, CGE generates more examples that represent novel domains (non-English languages and source code) in the fine-tuning dataset.

| iter | texts sampled by CGE from the fine-tuned OpenLlama |
|---|---|
| 1 | (Arabic) ... جان المان (17فبراير1950 -3يوليو1989)،مؤسسمصرواحةسابقةبكرة الليمون.منمؤسسات النضغة العمرانيةفي اليوم السابعفي21يوليوساهما الضرائبغيرأقل. |
| 2 | (Hebrew) ... היאאחתמ-46סוגימינההכוללתמחמישפעותבתחםקרהתابوربהשלהמינים("תנועהבשפההיוונת:באר"، "חֲטֻטֶר"، ו-פֻרוֹט، "בעלי'אדם |
| 3 | (Persian) ... حزبکمونیست اوندوبختیار (29ژانویه،1917–3ژانویه2004)کمونیستگونهایحزبی استکهدرقسمتغربیجهانبا انتخاباتزیادیودمکراترا انقلابیمیکرد. |
| 4 | (Thai) ... ซิเล็กตรอน ()เป็นเครื่องจากเล็กในบทเครื่องในกลองจักรและแว็ปไฮเดรนไกโตรี่ฟินแดงโดยเฉพาะนี่ |
| 5 | (Hebrew) ... מכונהבנושלרבמשחם,ממבקריכוחעצם.הואארבעמנימואצאאדם(Meshech :ריצה:מֶשֶׁ,חֶם;בלטינית) רבימְשׁוֹחֶמַמההתנ"ך |

| iter | texts randomly sampled from the fine-tuned OpenLlama |
|---|---|
| 1 | Home » Posts Tagged "Melbourne Cup"¥nRacing at Sandown Posted By admin on Oct1,2013 |0 comments ... (English) |
| 2 | The last time I saw my dad was on Christmas Eve in2017. We had a great day and he was happy, and I ... (English) |
| 3 | (Persian) ... زندگی .ریالدینرضازندی (زادۀ14سپتامبر1924 -درگذشته27دسامبر1976)شاعرونویسندهجغرافیایيمنتشرشد |
| 4 | This year the theme of the annual meeting of the European Society for Blood and Marrow Transplantation ... (English) |
| 5 | Home » News » Local » Gainesville » Gainesville police release names of victims in fatal shooting ... (English) |

Figure 4: Samples generated by OpenLLaMa fine-tuned on the non-English dataset. Texts generated for the first five iterations are shown. CGE generates more examples that represent novel domains (non-English languages) in the fine-tuning dataset.

| iter | texts sampled by CGE from the fine-tuned Falcon-RW |
|---|---|
| 1 | #!/bin/bash¥n¥n# Copyright 2016 The Kubernetes Authors All rights reserved.¥n#¥n# Licensed under the Apache License 2.0 (the "License"). You may not use this file except in compliance¥n# with the License. You can obtain a copy in the LICENSE file generated by this pipeline¥'s¥n# deploy script. Code generated by this pipeline¥'s deploy (Bash) |
| 2 | # TODO(john): Move this into a helper function so we can reuse this everywhere.¥nfunction cleanup() {¥n¥tkill ${KUBERNETES_USER:-} "$@";¥n¥tkill ${KUBERNETES_ADMIN:-} "$@";¥n¥tsleep 5;¥n¥tkill ${KUBERNETES_USER} "$@";¥n¥tsleep 2;¥n¥tsleep 5;¥n¥tkill ${KUBERNETES_ADMIN} "$@";¥n} (Bash) |
| 3 | __slots__ = [¥'model¥', ¥'category¥']¥n @property¥n def name(cls):¥n return cls¥n @property¥n def class(cls, meta):¥n return meta.__class__¥n @property¥n def name_v2(cls, meta, name_v2_type=¥'V2Class¥', name_v2_args=¥'V2ClassMetaKey, V2ClassMetaArgs¥', (Python) |
| 4 | import logging¥n¥nfrom chromite.common.logging import log_basic as log_basic¥n¥nclass Logger(object):¥n¥n def __init__(self):¥n self.name = 'chrome'¥n def __enter__(self, message):¥n super().enter()¥n¥n def __exit__(self, reason):¥n super().exit()¥n¥n¥nclass LoggerFactory(object):¥n¥n def __new__(self, (Python) |
| 5 | module Mongo¥n module Connection¥n class Client < ::Mongo::Connection¥n¥n attr_accessor :address, :port, :user, :password, :ssl, :ssl_cert_file, :ssl_cert_dir, :ssl_cert_path¥n¥n attr_accessor :timeout, :timeout_ms¥n¥n def initialize¥n @address = nil¥n (Ruby) |

| iter | texts randomly sampled from the fine-tuned Falcon-RW |
|---|---|
| 1 | The following lists events that happened during 1997 in New Zealand.¥n¥nIncumbents¥n¥nRegal and viceregal¥nHead of State – Elizabeth II¥nGovernor-General – The Rt Hon. Sir Michael Hardie Boys GNZM, GCMG, QSO¥n¥nGovernment¥nThe 44th New Zealand Parliament continued. The 47th New Zealand Parliament was to have been convened, but it was not ... (non-code) |
| 2 | The Battle of Long Island was a battle fought between the Continental Army and the British Army during the American Revolutionary War. The battle took place on Long Island, New York, on July 17, 1776, during the Siege of Boston.¥n¥nBackground¥n¥nOn June 17, 1776, the Continental Army under the command of General George ... (non-code) |
| 3 | The United States Coast Guard Academy (USCG) is a four-year military service academy located in New London, Connecticut. The academy is part of the United States Coast Guard and is responsible for the training of the Coast Guard's officers.¥n¥nHistory¥n¥nThe Coast Guard Academy was established as a result of the Coast Guard Act of 1920, ... (non-code) |
| 4 | /*¥n * Licensed to the Apache Software Foundation (ASF) under one¥n * or more contributor license agreements. See the NOTICE file¥n */¥n¥npackage org.apache.hadoop.hdfs;¥n¥nimport java.io.IOException;¥nimport java.io.InputStream;¥nimport java.io.InputStreamReader;¥nimport java.io.OutputStream;¥nimport java.io. ... (Java) |
| 5 | The Trans-Siberian Orchestra is a symphonic rock band formed in 2000 by Orianthi. The band consists of Orianthi (lead vocals, piano, keyboards, guitars), Michael Shrieve (guitars, backing vocals, keyboards, keyboards, programming, programming, bass, backing vocals) and Michael Shultz (drums, percussion, backing vocals, keyboards, programming, programming)... (non-code) |

Figure 5: Samples generated by Falcon fine-tuned on the source code dataset. Texts generated for the first five iterations are shown. CGE generates more examples that represent novel domains (source code) in the fine-tuning dataset.

