# OpenReview forum: "What's New in My Data? Novelty Exploration via Contrastive Generation"
_ICLR.cc/2025/Conference — ICLR 2025 Poster_

### Official Review · Reviewer_carY · 2024-11-04

**Soundness:** 3
**Presentation:** 2
**Contribution:** 2
**Rating:** 6
**Confidence:** 4

**Summary:**

This paper presents a method called Constrastive Egneration Exploration(CGE) and its interative version, which aims to capture novel characteristics of fine-tuning data. Since fine-tuning data might be confidential sometime, this work are something trying to generate examples that similar to fine-tuning data. More specificlly, by using probability of fine-tuned models minus probability of pre-trained models (called Contrastive Score), it apply contrastive decoding to generate examples. Experiments results show contrasrtive score is able to robustly identify such ood examples, and

**Strengths:**

1. The question itself seems new. Though using so called contrastive score (difference between two model's probability) to identify some ood texts is not something new, but successfully using it to somehow identify/recover part of fine-tuning datasets is intertesing.
2. It might be interesting to see whether this can be used as an attack method to recover some confidential fine-tuning data (given known pretrained model and fine-tune model.

**Weaknesses:**

1. The application of this question seems limited, because for real-world cases, i.e. confidential fine-tuning datasets, we might not know which pre-train model they are using. Also we may not be able to access the probability of fine-tuned model.
2. Experiments are limited and not so solid. Consider authors are using only two models on two different settings. (1) it is better to do controlled experiments. When you change to a new experimental setting, you can keep the model and other things unchanged. (2) The number of ood-domain is limited.
3. Writing about evaluation on novel examples is not clear.

**Questions:**

1. For the iterative version, how to make sure you fine-tune the pre-trained model on generated text can produce a new model with appropreate distribution. i.e. for the first iteration, the distribution of pre-trained model has no problem, and you identify and generate some  Japanese text. However, when you fine-tune on Japanese task, it is possible Japanese text will dominent the distribution and English/Russian will be "novel character" and it is possible to generate English again.
2. The evaluation process for different domains (i.e. Non-English and toxic) is not clear. Which domain is chosen as the ground truth for first iteration? How the accuracy is calculated based on the 100 generated texts? They are not clearly stated.

---

> ### Author Response · Authors · 2024-11-20
> **Response (1/2)**
>
> Thank you very much for your comments and feedback. We have revised the manuscript based on your feedback, with the changes highlighted in red color.
>
> ---
>
> > The application of this question seems limited, because for real-world cases, i.e. confidential fine-tuning datasets, we might not know which pre-train model they are using. Also we may not be able to access the probability of fine-tuned model.
>
> We address a scenario where model developers or analysts have access to both the pre-trained and fine-tuned models but do not have direct access to the fine-tuning dataset. This situation is common in industry, especially when subcontractors develop language models (LMs) using private or confidential data (see references [1, 2], where it is good practice that subcontractors are restricted to access the dataset ). For instance, developing a customer-care chatbot may involve fine-tuning on customer-service interaction data, to which developers are restricted from direct access. In these cases, insights about the dataset must be drawn indirectly from the model’s outputs.
>
> We would like to emphasize that our approach is easy to use and does not require any preprocessing (e.g., building an n-gram index or clustering), which is important from a practical perspective. This may make our approach a method of choice even in scenarios where access to the data is available.
>
> [1] Garrido et al., [Lessons learned: Surveying the practicality of differential privacy in the industry](https://arxiv.org/abs/2211.03898). Proceedings on Privacy Enhancing Technologies, 2023.
> [2] Sarathy et al., [Don’t look at the data! how differential privacy reconfigures the practices of data science](https://arxiv.org/abs/2302.11775). CHI, 2023.
>
> ---
>
>
> > Writing about evaluation on novel examples is not clear.
>
> > The evaluation process for different domains (i.e. Non-English and toxic) is not clear. Which domain is chosen as the ground truth for first iteration? How the accuracy is calculated based on the 100 generated texts? They are not clearly stated.
>
> We apologize for the lack of clarity in our initial explanation. For instance, in the non-English setting, we regard 10 non-English languages (e.g., Japanese, Chinese, Persian) as positive examples, with English text considered as negative examples. The detection rate is then calculated as the percentage of positive examples among the 100 generated texts. The coverage rate reflects the proportion of unique non-English languages represented in the generated texts, relative to the total distinct languages (10). For example, if 100 generated texts include 50 English, 30 Japanese, and 20 Chinese texts, the detection rate is 0.5 [(30+20)/100], and the coverage rate is 0.2 [2/10]. The coverage metric approximates the number of novel domains an analyst can detect within a limited time frame.  We clarified this in Section 4.3.
>
> ---
>
> > For the iterative version, how to make sure you fine-tune the pre-trained model on generated text can produce a new model with appropreate distribution. i.e. for the first iteration, the distribution of pre-trained model has no problem, and you identify and generate some Japanese text. However, when you fine-tune on Japanese task, it is possible Japanese text will dominent the distribution and English/Russian will be "novel character" and it is possible to generate English again.
>
> Yes, such situations can happen, and the iterative version generates English as “novel” examples. That’s why the detection rate substantially drops for the iterative version, as shown in Figure 2 (left). However, even if such cases happen, the iterative version tends to generate a wider variety of novel examples than the static version, as demonstrated by the higher coverage rate in Figure 2 (right).

---

> > ### Author Response · Authors · 2024-11-20
> > **Response (2/2)**
> >
> > > Experiments are limited and not so solid. Consider authors are using only two models on two different settings. (1) it is better to do controlled experiments. When you change to a new experimental setting, you can keep the model and other things unchanged. (2) The number of ood-domain is limited.
> >
> > In response, we have expanded our experiments by including a source code dataset in the OpenLLaMA experiment and a toxic text dataset in the Falcon-RW experiment (Table 1 and 2). We also added recent novelty detection methods, Entropy [1] and GEN [1], to the experiments (Table 1).
> >
> > In Table 1, the scoring function consistently shows strong performance across both settings (AUROC is above 0.9 for both cases), while the added baseline methods do not perform well specifically for non-English and source code datasets.
> >
> > ・Results on the extraction setup for OpenLLaMA (Table 1):
> > |                   |AUROC (non-en)|FPR95 (non-en)|     |AUROC (code)|FPR95 (code)|     |AUROC (toxic)|FPR95 (toxic)|
> > |:------------------|--------------:|--------------:|:----|-------------:|-------------:|:----|--------:|--------:|
> > | Entropy           |          0.10  |          1.00    |     |         0.06 |         1.00    |     |    0.96 |    0.25 |
> > | GEN               |          0.14 |          1.00    |     |         0.19 |         0.98 |     |    1.00    |    0.03 |
> > | Contrastive score |          0.99 |          0.05 |     |         0.92 |         0.24 |     |    0.95 |    0.34 |
> >
> > ・Results on the extraction setup for Falcon-RW (Table 1):
> > |                   |AUROC (non-en)|FPR95 (non-en)|     |AUROC (code)|FPR95 (code)|     |AUROC (toxic)|FPR95 (toxic)|
> > |:------------------|--------------:|--------------:|:----|-------------:|-------------:|:----|--------:|--------:|
> > | Entropy           |          0.20  |          0.89 |     |         0.05 |         1.00    |     |     1.00   |    0.00    |
> > | GEN               |          0.16 |          0.90  |     |         0.33 |         0.92 |     |     1.00   |    0.00    |
> > | Contrastive score |          0.98 |          0.11 |     |         0.93 |         0.13 |     |     0.90 |    0.92 |
> >
> > In Table 2, CGE (static) achieves the highest detection rate for the toxic text dataset of Falcon-RW, while CGE (iterative) attains the highest coverage rate for the source code dataset of OpenLLaMA.
> >
> > ・Results on the generation setup for OpenLLaMA (Table 2):
> > |                 |Detection (non-en)|Coverage (non-en)|     |Detection (code)|Coverage (code)|     |Detection (toxic)|Coverage (toxic)|
> > |:--------------------|:------------------|:-----------------|:----|:-----------------|:----------------|:----|:------------|:-----------|
> > | Sampling        | 0.28±0.03         | 0.65±0.05        |     | 0.56±0.06        | 0.90±0.07       |     | 0.33±0.05   | 0.93±0.08  |
> > | CGE (static)    | 0.99±0.01         | 0.55±0.05        |     | 0.31±0.06        | 0.53±0.13       |     | 0.78±0.06   | 0.75±0.05  |
> > | CGE (iter) | 0.18±0.03         | 0.82±0.11        |     | 0.28±0.01        | 0.97±0.04       |     | 0.35±0.03   | 0.83±0.08  |
> >
> > ・Results on the generation setup for Falcon-RW (Table 2):
> > |                 |Detection (non-en)|Coverage (non-en)|     |Detection (code)|Coverage (code)|     |Detection (toxic)|Coverage (toxic)|
> > |:--------------------|:------------------|:-----------------|:----|:-----------------|:----------------|:----|:------------|:-----------|
> > | Sampling        | 0.01±0.00         | 0.12±0.04        |     | 0.06±0.03        | 0.25±0.09       |     | 0.12±0.02   | 0.55±0.09  |
> > | CGE (static)    | 0.53±0.05         | 0.43±0.04        |     | 0.92±0.02        | 0.62±0.04       |     | 0.36±0.03   | 0.75±0.11  |
> > | CGE (iter) | 0.14±0.08         | 0.55±0.11        |     | 0.42±0.17        | 0.90±0.07       |     | 0.09±0.04   | 0.65±0.26  |
> >
> >
> > [1] Liu et al., [Gen: Pushing the limits of softmax-based out-of-distribution detection](https://openaccess.thecvf.com/content/CVPR2023/papers/Liu_GEN_Pushing_the_Limits_of_Softmax-Based_Out-of-Distribution_Detection_CVPR_2023_paper.pdf). CVPR, 2023.

---

> > > ### Author Response · Authors · 2024-11-23
> > >
> > > Dear Reviewer carY,
> > >
> > > As we approach the end of this discussion period, we want to thank you for your thoughtful feedback, which has significantly improved our paper. We believe we’ve addressed most of the concerns raised by you and the other reviewers. Specifically, we have revised our paper as follows:
> > >
> > > - We have elaborated the application of our problem settings in the introduction. In industry, developers commonly have access to both pre-trained and fine-tuned models but not the fine-tuning dataset. For instance, subcontractors develop a customer-care chatbot on customer-service interaction data, which are restricted from direct access.
> > > - We clarified the evaluation metrics for the experiments in Section 4.3. For example, in the non-English setting, the detection rate represents the proportion of positive examples (e.g., non-English texts) among the generated texts, while the coverage rate reflects the fraction of unique novel domains detected. These metrics assess the variety and relevance of the identified novel examples.
> > > - We expanded the experiments by including a source code dataset for OpenLLaMA and a toxic text dataset for Falcon-RW (Tables 1 & 2), as well as additional novelty detection baselines (Entropy and GEN) in Table 1. The scoring function remains robust across diverse datasets, outperforming baselines on the non-English and source code datasets.
> > >
> > > We would appreciate it if you could let us know whether our responses properly address your concerns.

---

> > > > ### Comment · Reviewer_carY · 2024-11-26
> > > > **Response**
> > > >
> > > > Dear authors,
> > > >  I am appreciated that author's explanation and revised manuscript to address my concern. After checking your revised manuscript and other reviewer's discussion:
> > > > 1. I understand your proposed scenario "where model developers or analysts have access to both the pre-trained and fine-tuned models but do not have direct access to the fine-tuning dataset". It is very common that developers have no access to the fine-tuned data, however, how many of them have access to the pre-trained model? Consider some may use closed-source models such as GPT-3/4 for fine-tuning, some may use Llama/Mistral/Qwen for further finetuning, you may not know exactly which pre-trained model they are using. Also for some situation, even fine-tuned models are not directly accessable, instead with API service. So I think this narrow the application of this method in real-world scenarios.
> > > > 2. The evaluation part is more clearly now, which helps us know what detection rate and coverage rate exactly is.
> > > > 3. If I understand correctly, the final coverage rate matters while final detection rate is not important right? so after several iteration, though it is possible to generate english text, but most of the novel domains has already be detected.
> > > >
> > > > I have changed my score accordingly.

---

> > > > > ### Author Response · Authors · 2024-11-26
> > > > >
> > > > > Dear Reviewer carY,
> > > > >
> > > > > Thank you very much for updating the score. We appreciate that the reviewer took the time to read our improved version of the manuscript and responses to other reviewers.
> > > > >
> > > > > > I understand your proposed scenario “where model developers or analysts have access to both the pre-trained and fine-tuned models but do not have direct access to the fine-tuning dataset”. It is very common that developers have no access to the fine-tuned data, however, how many of them have access to the pre-trained model? Consider some may use closed-source models such as GPT-3/4 for fine-tuning, some may use Llama/Mistral/Qwen for further finetuning, you may not know exactly which pre-trained model they are using. Also for some situation, even fine-tuned models are not directly accessable, instead with API service. So I think this narrow the application of this method in real-world scenarios.
> > > > >
> > > > > We acknowledge that our method is not applicable in scenarios where access to pre-trained or fine-tuned models is restricted, such as in API-only interactions (e.g., with OpenAI models).  In contrast, for cases where subcontractors/developers work with models like Llama or Mistral, they generally select the pre-trained model to fine-tune and, thus, are aware of the model used during fine-tuning. However, we acknowledge that there are situations where such access may be unavailable, or where data analysts may lack knowledge of the pre-trained model employed for fine-tuning.
> > > > >
> > > > > > If I understand correctly, the final coverage rate matters while final detection rate is not important right? so after several iteration, though it is possible to generate english text, but most of the novel domains has already be detected.
> > > > >
> > > > > Yes, coverage rate is more critical as it reflects the number of novel domains detected. A higher coverage rate with a lower detection rate means that the analysts will lose some time reviewing non-novel examples but will uncover a broader range of novel domains, which aligns with the primary goal of our method.

---

### Official Review · Reviewer_rHvj · 2024-11-05

**Soundness:** 2
**Presentation:** 2
**Contribution:** 2
**Rating:** 6
**Confidence:** 3

**Summary:**

The authors propose a method based on contrastive learning that can generate examples from classes that are in the fine tuning set that are not in the pre-training set. The authors' experiments involves evaluating GradNorm and various proposed probabilities methods on OpenLLaMA and Falcon-RW.

**Strengths:**

- Identifying what kind of examples in the fine-tuning set that are not present in the pretraining set is interesting.

**Weaknesses:**

- The writing is poor, and the story and motivation are vague and confusing. The authors keep switching between writing "novelty exploration", "novel characteristics", and "novel domains" which makes it difficult to understand what the actual task is. After much digging, this work is basically about identifying example classes that are in the fine-tuning set and not in the pretraining set. Perhaps the title could be, "identifying new classes in the finetuning set".

- It is not clear what the goal of this work is in terms of the benefits of identifying new classes in the fine-tuning set. The introduction mentions that this capability would allow us to remove toxic values, but how would the model know the new classes are "toxic"?

- There are so many trivial ways to identify whether the model knows whether a certain type of examples were in the pretraining set. What if you simply compute the uncertainty or the entropy of the model output on the examples, and identify those with the highest entropy? those are most likely examples that the model hasn't seen.

**Questions:**

Could you address the weaknesses above?

---

> ### Author Response · Authors · 2024-11-20
>
> Thank you very much for your comments and feedback. We have revised the manuscript based on your feedback, with the changes highlighted in red color.
>
> ---
>
> > The writing is poor, and the story and motivation are vague and confusing. The authors keep switching between writing "novelty exploration", "novel characteristics", and "novel domains" which makes it difficult to understand what the actual task is. After much digging, this work is basically about identifying example classes that are in the fine-tuning set and not in the pretraining set. Perhaps the title could be, "identifying new classes in the finetuning set".
>
> Thank you for the helpful suggestions. We have revised the paper to use the term “novel domain” consistently throughout the manuscript, to clarify the task. We also edited the introduction to clarify the motivation.
>
> ---
>
> > It is not clear what the goal of this work is in terms of the benefits of identifying new classes in the fine-tuning set. The introduction mentions that this capability would allow us to remove toxic values, but how would the model know the new classes are "toxic"?
>
>
> The goal of this work is not to create a fully automated system but rather to equip analysts or developers with a tool that enables them to discover new domains within the data (without direct access to the data). If they detect an undesirable domains, they can take corrective actions, such as adding filters, performing model edits, or even deciding that the model poses too great a risk to deploy. We edited the introduction to emphasize this scenario.
>
> ---
>
> > There are so many trivial ways to identify whether the model knows whether a certain type of examples were in the pretraining set. What if you simply compute the uncertainty or the entropy of the model output on the examples, and identify those with the highest entropy? those are most likely examples that the model hasn't seen.
>
> Our primary goal in this work is the discovery of new domains using a fine-tuned model, which precludes the use of simpler methods, including those you proposed. We also do not assume that the analyst have access to the data or has a preconception of what specific novel domains (e.g., type of toxic data) may be present.
>
> However, in Section 4.2 (rather than in the main experiments in Section 4.3), we assume access to the fine-tuning dataset and perform sentence selection to showcase the scoring function's effectiveness. Additionally, we have incorporated your proposed method as a baseline in Section 4.2 (see Table 1), following a recent work that uses Shannon entropy and generalized entropy for ood detection [1]. Though they work well in detecting toxic text, they do not perform well specifically for non-English and source code datasets because the entropy of non-English text and source code is not generally high (more details are described in Section 4.2). In contrast, the contrastive score focuses on the difference in the log probability rather than their absolute values, performing robustly across various types of novelties.
>
> [1] Liu et al., [Gen: Pushing the limits of softmax-based out-of-distribution detection](https://openaccess.thecvf.com/content/CVPR2023/papers/Liu_GEN_Pushing_the_Limits_of_Softmax-Based_Out-of-Distribution_Detection_CVPR_2023_paper.pdf). CVPR, 2023.
>
> ・Results for OpenLLaMA (Table 1):
> |                   |AUROC (non-en)|FPR95 (non-en)|     |AUROC (code)|FPR95 (code)|     |AUROC (toxic)|FPR95 (toxic)|
> |:------------------|--------------:|--------------:|:----|-------------:|-------------:|:----|--------:|--------:|
> | Entropy           |          0.10  |          1.00    |     |         0.06 |         1.00    |     |    0.96 |    0.25 |
> | GEN               |          0.14 |          1.00    |     |         0.19 |         0.98 |     |    1.00    |    0.03 |
> | Contrastive score |          0.99 |          0.05 |     |         0.92 |         0.24 |     |    0.95 |    0.34 |
>
> ・Results for Falcon-RW (Table 1):
> |                   |AUROC (non-en)|FPR95 (non-en)|     |AUROC (code)|FPR95 (code)|     |AUROC (toxic)|FPR95 (toxic)|
> |:------------------|--------------:|--------------:|:----|-------------:|-------------:|:----|--------:|--------:|
> | Entropy           |          0.20  |          0.89 |     |         0.05 |         1.00    |     |     1.00   |    0.00    |
> | GEN               |          0.16 |          0.90  |     |         0.33 |         0.92 |     |     1.00   |    0.00    |
> | Contrastive score |          0.98 |          0.11 |     |         0.93 |         0.13 |     |     0.90 |    0.92 |

---

> > ### Author Response · Authors · 2024-11-23
> >
> > Dear Reviewer rHvj,
> >
> > As we approach the end of this discussion period, we want to thank you for your thoughtful feedback, which has significantly improved our paper. We believe we’ve addressed most of the concerns raised by you and the other reviewers. Specifically, we have revised our paper as follows:
> >
> > - We have consistently used the term "novel domain" throughout the paper and revised the introduction to clarify the motivation and task.
> > - We have clarified the goal and benefits of our problem setting. This work is not intended to create a fully automated system but rather to provide analysts or developers with a tool for discovering novel domains within fine-tuning datasets, even without direct dataset access.
> > - We have elaborated on our problem setting, which assumes that the pre-training and fine-tuning datasets are not accessible, and previous novelty detection methods (e.g., entropy-based approaches) cannot be applicable to this scenario (Sections 1 & 2).
> > - In the experimental setting where we have access to the fine-tuning datasets, we have included the suggested method as baselines (Entropy and Gen in Table 1). We found they work well for toxic text but perform poorly for non-English and source code datasets due to the characteristics of the datasets. Our contrastive score, focusing on differences in log probabilities, demonstrates robust performance across various novel domains.
> >
> > We would appreciate it if you could let us know whether our responses properly address your concerns.

---

> > > ### Author Response · Authors · 2024-11-29
> > >
> > > Dear Reviewer rHvj,
> > >
> > > Following up on our previous reminder, we wanted to let you know that we have made substantial revisions to the paper, including new experiments aimed at addressing your concerns (e.g., additional baselines and clarification of our problem setup as detailed in our previous messages). We would appreciate your feedback on whether these updates address your critiques.

---

> > > > ### Comment · Reviewer_rHvj · 2024-12-02
> > > >
> > > > Thank you for your response. Most of the comments are clear and helpful, but the writing could still be improved, and the goal needs to be more clearly motivated. I raise my score to a 6.

---

> > > > > ### Author Response · Authors · 2024-12-02
> > > > >
> > > > > Reviewer rHvj,
> > > > >
> > > > > Thank you for your feedback and raising the score. We’ll make sure to further clarify the goals in the next revision. We appreciate the time and effort you (and the other reviewers) put into this -- it’s made the paper much stronger.
> > > > >
> > > > > Best regards,
> > > > > the authors

---

### Official Review · Reviewer_J5Ry · 2024-11-06

**Soundness:** 3
**Presentation:** 3
**Contribution:** 3
**Rating:** 6
**Confidence:** 4

**Summary:**

In this paper, the authors introduce to discover through generation, which aims to identify properties in a fine-tuning dataset without directly accessing the fine-tuning data. They propose the Contrastive Generative Exploration (CGE) to fine-tune datasets by generating examples that represent these properties.

**Strengths:**

1. The experimental setting are soild. They conduct experiments on multiple base LLM and mulitple datasets. The authors report the experimental results on multiple evaluation metrics.

2. To achieve contrastive generative exploration, the methods consists of static approach, iterative approach, and so on. The design of methods seem reasonable.

3. The task (direction) this paper focusing on sounds interesting. Achieve a indirect and effective tine-tuning is curical in LLM and generalized methods can be widely-used in many LLM scenarios.

**Weaknesses:**

1. The baseline methods are kind of old and weak. It would be better to compare methods proposed in 2023 or 2024, but the baselines used in this submission are proposed before 2021.

2. To compare with DP-based methods, the stardard setting is to compare over different \epsilon instead of the noise multiplier. \epsilon indicates the strength of the privacy protection but noise multiplier is not so clear to show the privacy protection performance since the scale of noises varies in different models.

3. The source code should be public accessible.

4. It is better to bold the best results in Table1. From the experimental results shown in Tables 1, the proposed model cannot always outperform the baselines' results. More detailed and insightful explanations are required.

**Questions:**

Will the authors release the code?

---

> ### Author Response · Authors · 2024-11-20
>
> Thank you very much for your comments and feedback. We have revised the manuscript based on your feedback, with the changes highlighted in red color.
>
> ---
> > The baseline methods are kind of old and weak. It would be better to compare methods proposed in 2023 or 2024, but the baselines used in this submission are proposed before 2021.
>
> The primary focus of our paper is to propose a method that can be used when the fine-tuning dataset is unavailable, and examples need to be generated rather than selected from a dataset (Section 4.3, Table 2). For this setting, there is no prior work, hence we can only use simple ablated versions of the model. We believe that this assumption matches practical setups and is important; introducing this setting is also a contribution of this paper. Moreover, our approach is very simple to use and does not require any time-consuming processing. Thus, it may be preferable even if the access is available, but the extra time necessary to build an index or process the dataset is undesirable.
>
> In Section 4.2, we show that the scoring function underlying our generative approach is sufficiently strong. The goal is to just show that it is robust across various settings, and performs better or on par than simple alternatives. Still, to address your reservation, we added several recent baseline methods, Entropy[1] and GEN [1] in Table 1 (Section 4.2). They employ Shannon entropy and generalized entropy for detecting OOD examples. Though they outperform other methods in their work, they do not perform well in our case, specifically for non-English and source code datasets. This is because the entropy of non-English text and source code is not generally high (details are described in Section 4.2). In contrast, the contrastive score focuses on the difference in the log probability rather than their absolute values, performing robustly across various types of novelties.
>
> [1] Liu et al., Gen: Pushing the limits of softmax-based out-of-distribution detection. CVPR, 2023.
>
> ・Results for OpenLLaMA (Table 1):
> |                   |AUROC (non-en)|FPR95 (non-en)|     |AUROC (code)|FPR95 (code)|     |AUROC (toxic)|FPR95 (toxic)|
> |:------------------|--------------:|--------------:|:----|-------------:|-------------:|:----|--------:|--------:|
> | Entropy           |          0.10  |          1.00    |     |         0.06 |         1.00    |     |    0.96 |    0.25 |
> | GEN               |          0.14 |          1.00    |     |         0.19 |         0.98 |     |    1.00    |    0.03 |
> | Contrastive score |          0.99 |          0.05 |     |         0.92 |         0.24 |     |    0.95 |    0.34 |
>
> ・Results for Falcon-RW (Table 1):
> |                   |AUROC (non-en)|FPR95 (non-en)|     |AUROC (code)|FPR95 (code)|     |AUROC (toxic)|FPR95 (toxic)|
> |:------------------|--------------:|--------------:|:----|-------------:|-------------:|:----|--------:|--------:|
> | Entropy           |          0.20  |          0.89 |     |         0.05 |         1.00    |     |     1.00   |    0.00    |
> | GEN               |          0.16 |          0.90  |     |         0.33 |         0.92 |     |     1.00   |    0.00    |
> | Contrastive score |          0.98 |          0.11 |     |         0.93 |         0.13 |     |     0.90 |    0.92 |
>
> ---
>
> > To compare with DP-based methods, the stardard setting is to compare over different \epsilon instead of the noise multiplier. \epsilon indicates the strength of the privacy protection but noise multiplier is not so clear to show the privacy protection performance since the scale of noises varies in different models.
>
> We have added the corresponding \epsilon values for each noise multiplier in Figure 3. The \epsilon values are the same for both OpenLLaMA and Falcon, as we used the same size of the datasets for fine-tuning these two models.
>
> ---
>
> > The source code should be public accessible.
>
> We made the code publicly available at the following link: https://anonymous.4open.science/r/cge/
>
> ---
>
> > It is better to bold the best results in Table1. From the experimental results shown in Tables 1, the proposed model cannot always outperform the baselines' results. More detailed and insightful explanations are required.
>
> Thank you for your feedback, we’ve bolded the best results as requested. The primary focus of our paper is to propose a method that can be used when the fine-tuning dataset is unavailable, and examples need to be generated rather than extracted. Table 1 presents preliminary experimentation that the scoring function underlying our contrastive generation method is accurate. In Section 4.3, the scoring function consistently shows strong performance across different models and datasets (AUROC is above 0.9 for all cases, and FPR95 is around 0.1 in 3 out of 6 cases). More importantly, the rest of the methods in Section 4.2 are not applicable to generation, i.e. the main focus of the paper (Section 4.3). We rephrased the introduction to Section 4.2 to clarify this.

---

> > ### Author Response · Authors · 2024-11-23
> >
> > Dear Reviewer J5Ry,
> >
> > As we approach the end of this discussion period, we want to thank you for your thoughtful feedback, which has significantly improved our paper. We believe we’ve addressed most of the concerns raised by you and the other reviewers. Specifically, we have revised our paper as follows:
> >
> > - We clarified our problem setting, which focuses on the scenario where the pre-training and fine-tuning datasets are not available. As previous ood detection methods cannot be applicable to this scenario, we employed simple ablated versions of our method. (Section 4.3)
> > - We demonstrated that our method robustly performs even in the setting where we have access to the fine-tuning dataset. We have added recent novelty detection methods proposed in CVPR 2023 (Entropy and Gen) to the experiment. The results demonstrated that these methods do not perform well on some datasets (Table 1 in Section 4.2). We also have bolded the best results in the table.
> > - We have made the code publicly available at the following link: https://anonymous.4open.science/r/cge/
> > - We have added the corresponding $\epsilon$ values for each noise multiplier to Figure 3.
> >
> > We would appreciate it if you could let us know whether our responses properly address your concerns.

---

> ### Author Response · Authors · 2024-11-29
>
> Dear Reviewer J5Ry,
>
> Following up on our previous reminder, we wanted to let you know that we have made substantial revisions to the paper, including new experiments aimed at addressing your concerns (e.g., additional baselines and code release as detailed in our previous messages). We would appreciate your feedback on whether these updates address your critiques.

---

> > ### Author Response · Authors · 2024-12-02
> >
> > Dear Reviewer J5Ry,
> >
> > This is a final reminder that today is the last day the reviewers can post messages on the forum. We understand it’s a busy time, but we’d greatly appreciate hearing your thoughts on the revised paper and responses.
> >
> > We’ve made significant updates, including new experiments and released code, to address your and other reviewers’ concerns.
> >
> > Thank you for your time,
> > the authors.

---

### Official Review · Reviewer_B4zm · 2024-11-07

**Soundness:** 3
**Presentation:** 2
**Contribution:** 2
**Rating:** 6
**Confidence:** 4

**Summary:**

This paper considers a novel task of finding what is new in a fine-tuning dataset, given the fact that we have a pre-trained model, but it is unclear what the distribution of data was for the same. In particular, they introduce Contrastive Generative Exploration or CGE, which tries to find novel properties which are unique to the fine-tuning data. This basically leverages the difference in predictions between the pre-trained model and the model fine-tuned on the new dataset, and appropriately generates new sentences. Using this metric itself of new probability scoring, they are able to get much higher true positive rates of detecting novel content. At the same time, they are also able to generate novel sentences. The work discusses two different methods of CGE, one which is iterative and one which is static, and do experiments on non-English languages or toxic text or code data to find novel content. Finally, there is also a section in the paper that talks about how this can work with differential privacy as well, and in scenarios when we do not have access to visualizing the true data.

**Strengths:**

- Shifting Focus from Data Inspection to Model Behavior: This is a neat transition because direct data access is often restricted due to privacy concerns or the sheer scale of the datasets involved. CGE, therefore, offers a practical and potentially more scalable solution for understanding novelties in fine-tuning data.
- Contrastive Decoding for Dataset Exploration: This technique, originally developed for controlling text generation properties, is cleverly adapted to distinguish between in-distribution and novel examples by contrasting the predictions of pre-trained and fine-tuned models.
- Synergy with Differentially Private fine-tuning: In line with the first goal of exploring datasets novelties without direct access to them, the paper does a good job at showing how a model fine-tuned with DP can still allow inspection of properties of the dataset it was trained on by looking at the induced model behaviors.
- High AUC with metric: The new metric of detecting novel content achieves substantially better performance than most existing metrics.

**Weaknesses:**

- Missing an honest baseline like n-gram: This paper misses basic baselines such as n-gram distribution change, or top-k nearest neighbour distance in the embedding space.
   - This could quite naturally be done in the RPJ experiments.
   - Can also be adapted in the DP-SGD experiments by returning privatized n-gram stats.
   - You might like the infini-gram paper as a tool to enable such analysis
- Constraint: The paper writes that one of the constraints for the method to work effectively is that the number of novel examples substantially smaller than the remaining examples in the fine-tuning data. This is a very hard-to-match constraint in my opinion
- No axis labels on graphs 2,3
- No samples shown:

Overall, my main concern with this work is that I am unable to position its applicability in the present day LLM use.

**Questions:**

- Can you think of adapting this metric to measure the distance between two distributions?
- I would like to view CDE generated samples

---

> ### Author Response · Authors · 2024-11-20
> **Response (1/2)**
>
> Thank you very much for your comments and feedback. We have revised the manuscript based on your feedback, with the changes highlighted in red color.
>
> ---
> > Missing an honest baseline like n-gram: This paper misses basic baselines such as n-gram distribution change, or top-k nearest neighbour distance in the embedding space.
> > ・This could quite naturally be done in the RPJ experiments.
> > ・Can also be adapted in the DP-SGD experiments by returning privatized n-gram stats.
> > ・You might like the infini-gram paper as a tool to enable such analysis
>
> Both these baseline methods (n-gram distribution changes and top-k nearest neighbor distance) assume access to the fine-tuning and pre-training datasets. These are assumptions that we did not make in our method.
>
> If direct access to the pre-training data (or its index) were available, we agree that this can be advantageous.  For most models, however, this access is unavailable. To see what happens when both pre-training and fine-tuning datasets are available, we ran the top-k nearest-neighbor distance measures [1] and showed its performance in Appendix A.4 (Table 5). K-NN performs exceptionally well for non-English text and source code, though it performs worse on the toxic text dataset for Falcon-RW. When both novel and in-distribution examples are available, distinguishing between them becomes trivial since typical English text and novel-domain texts (e.g., non-English text, programming languages, and toxic text) differ significantly, and the latent representations effectively capture these differences.
>
> However, we relax the assumption of having no access to the fine-tuning dataset in Section 4.2. This serves two purposes: it allows us to evaluate the underlying scoring function - the difference in the log probability - in a simpler setup (before using it in generation in Section 4.3), and also to show that our method is competitive even in this setting. Now, we added recent novelty detection methods, Entropy [2] and GEN [2], to these experiments. These methods employ the Shannon entropy and generalized entropy of the pre-trained model’s predictions and do not assume access to the pre-training dataset. As shown in Table 1, they do not perform well in our case, specifically for non-English and source code datasets, because the entropy of non-English texts and source code is generally low (details are described in Section 4.2). In contrast, the contrastive score focuses on the difference in the log probability rather than their absolute values, performing robustly across various types of novelties.
>
> We would like to emphasize that our approach is easy to use and does not require any preprocessing (e.g., building an n-gram index or clustering), which is important from a practical perspective. This may make our approach a method of choice even in scenarios where access to the data is available.
>
>
> [1] Sun et al., [Out-of-Distribution Detection with Deep Nearest Neighbors](https://arxiv.org/abs/2204.06507). ICML, 2022.
> [2] Liu et al., [Gen: Pushing the limits of softmax-based out-of-distribution detection](https://openaccess.thecvf.com/content/CVPR2023/papers/Liu_GEN_Pushing_the_Limits_of_Softmax-Based_Out-of-Distribution_Detection_CVPR_2023_paper.pdf). CVPR, 2023.
>
> ・Results for OpenLLaMA (Table 1):
> |                   |AUROC (non-en)|FPR95 (non-en)|     |AUROC (code)|FPR95 (code)|     |AUROC (toxic)|FPR95 (toxic)|
> |:------------------|--------------:|--------------:|:----|-------------:|-------------:|:----|--------:|--------:|
> | Entropy           |          0.10  |          1.00    |     |         0.06 |         1.00    |     |    0.96 |    0.25 |
> | GEN               |          0.14 |          1.00    |     |         0.19 |         0.98 |     |    1.00    |    0.03 |
> | Contrastive score |          0.99 |          0.05 |     |         0.92 |         0.24 |     |    0.95 |    0.34 |
>
> ・Results for Falcon-RW (Table 1):
> |                   |AUROC (non-en)|FPR95 (non-en)|     |AUROC (code)|FPR95 (code)|     |AUROC (toxic)|FPR95 (toxic)|
> |:------------------|--------------:|--------------:|:----|-------------:|-------------:|:----|--------:|--------:|
> | Entropy           |          0.20  |          0.89 |     |         0.05 |         1.00    |     |     1.00   |    0.00    |
> | GEN               |          0.16 |          0.90  |     |         0.33 |         0.92 |     |     1.00   |    0.00    |
> | Contrastive score |          0.98 |          0.11 |     |         0.93 |         0.13 |     |     0.90 |    0.92 |

---

> ### Author Response · Authors · 2024-11-20
> **Response (2/2)**
>
> ---
>
> > Constraint: The paper writes that one of the constraints for the method to work effectively is that the number of novel examples substantially smaller than the remaining examples in the fine-tuning data. This is a very hard-to-match constraint in my opinion
>
> It was not our assumption, and we rephrased the text for clarity (see Section 2). Our model does demonstrate strong performance when the proportion of novel examples is high (see Table 4 in Appendix A.3). Our objective in experiments was to show that our method remains effective even in the challenging scenario where certain novel domains are ‘hidden’ (e.g., cannot be at all surfaced through random sampling from fine-tuned models). The ability to reveal such hidden domains is crucial, as harmful data - such as toxic content or misinformation - may present in training in tiny amounts yet still compromise model safety.
>
> ・Performance of CGE when varying the proportion of novel examples in the fine-tuning dataset (Table 4):
> |                 |Detection (10%)|Coverage (10%)|    |Detection (50%)|Coverage (50%)|     |Detection (90%)|Coverage (90%)|
> |:----------------|:------------|:-----------|:----|:--------------|:-------------|:----|:--------------|:-------------|
> | Sampling        | 0.01±0.00   | 0.12±0.04  |     | 0.93±0.03     | 0.77±0.08    |     | 1.00±0.00     | 0.62±0.08    |
> | CGE (static)    | 0.53±0.05   | 0.43±0.04  |     | 1.00±0.00     | 0.45±0.09    |     | 1.00±0.00     | 0.47±0.08    |
> | CGE (iter) | 0.14±0.08   | 0.55±0.11  |     | 0.77±0.03     | 0.95±0.05    |     | 0.97±0.01     | 1.00±0.00    |
>
> ---
>
> > No [y-]axis labels on graphs 2,3
>
> The titles shown about the graphs are also the y-axis labels. To prevent any confusion, we have annotated y-axis labels explicitly.
>
> ---
>
> > No samples shown.
>
> We have included examples generated by our method in Appendix A.6 (Figure 4 and 5).
> We also made the code publicly available at the following link: https://anonymous.4open.science/r/cge/
>
> ---
>
> > Can you think of adapting this metric to measure the distance between two distributions?
>
> This is an interesting question. Intuitively, using importance sampling with samples from the renormalized distribution q = renorm(p_ft / p_pt) could serve as an effective proposal for estimating KL(p_ft || p_pt) (or any f-divergence more generally), as it emphasizes examples that contribute significantly to the KL divergence. However, the connection may be less straightforward, as our decoding method does not directly produce samples from q.

---

> ### Author Response · Authors · 2024-11-23
>
> Dear Reviewer B4zm,
>
> As we approach the end of this discussion period, we want to thank you for your thoughtful feedback, which has significantly improved our paper. We believe we’ve addressed most of the concerns raised by you and the other reviewers. Specifically, we have revised our paper as follows:
>
> - We have clarified our problem setting, which focuses on the scenario where the pre-training and fine-tuning datasets are not available, and previous novelty detection methods (e.g., kNN and n-gram distribution changes) are not applicable to this scenario (Section 1 & 2).
> - We demonstrated that our method robustly performs even in the setting where we have access to the fine-tuning dataset. We have added recent novelty detection methods (Entropy and Gen) to the experiments, where they do not perform well on some datasets (Table 1 in Section 4.2).
> - We clarified that our method does not require the assumption that the number of novel examples is smaller than that of in-distribution examples in the fine-tuning dataset. We showed our method performs well even when the majority of the dataset consists of novel examples (Table 4 in Appendix A.3).
> - We have added y-axis labels to the graphs and included examples generated by our method. (Figures 4 and 5 in Appendix A.6)
>
> We would appreciate it if you could let us know whether our responses properly address your concerns.

---

> > ### Author Response · Authors · 2024-11-26
> >
> > Dear Reviewer B4zm,
> >
> > We sincerely apologize for following up with another reminder, but we wished to kindly bring to your attention that we have made substantial revisions to the paper, including new experiments aimed at addressing your concerns (as detailed in our previous messages). We would greatly appreciate your thoughts on whether these updates adequately address your critiques.

---

> > > ### Author Response · Authors · 2024-12-02
> > >
> > > Dear Reviewer B4zm,
> > >
> > > This is a final reminder that today is the last day the reviewers can post messages on the forum. We understand it’s a busy time, but we’d greatly appreciate hearing your thoughts on the revised paper and responses.
> > >
> > > We’ve made significant updates, including new experiments, to address your and other reviewers’ concerns. The other reviewers have shared positive feedback on the revisions, and your feedback would also be invaluable.
> > >
> > > Thank you for your time,
> > > Authors.

---

> > > > ### Comment · Reviewer_B4zm · 2024-12-03
> > > >
> > > > Dear Authors,
> > > >
> > > > First, I apologize for the delayed response to your follow-up messages. Let me share my thoughts based on reading your responses and the revised manuscript.
> > > >
> > > > **Baseline Comparisons and Method Requirements**
> > > > Your clarification about the distinction between methods requiring dataset access versus your approach is well-taken. The addition of Entropy and GEN baselines significantly strengthens your evaluation. However, I suggest:
> > > > - Adding a clear comparison table in the main paper showing which methods require what types of access (data, model, etc.)
> > > > - Elaborating on the practical implications of not requiring preprocessing and a brief discussion of trade-offs between performance and access requirements
> > > >
> > > > **Novel Examples and Scalability**
> > > > I appreciate the clarification about performance with varying proportions of novel examples. The results in Table 4 are particularly compelling. To make this even clearer:
> > > > - Consider moving Table 4 to the main paper as it addresses a key concern about practical applicability
> > > > - Add a discussion about how performance characteristics change across different proportions
> > > >
> > > > **Visualization and Example Analysis**
> > > > Thanks for the improvements here. As a part of reproducibility and access, you could also release a Huggingface dataset of "novelties" to make viewing convenient
> > > >
> > > >
> > > > **Overall Assessment**
> > > > The revisions have substantially improved the paper's clarity and technical depth. While some concerns remain about practical applicability in current LLM workflows, your additional experiments and clarifications have helped position the work more clearly in the field. I appreciate your diligence in addressing the review comments and the thoroughness of your responses. I am raising my score from 3 to 6.

---

> ### Author Response · Authors · 2024-12-03
>
> Dear Reviewer B4zm,
>
> Thank you so much for your feedback and for raising your score.
> We appreciate your recognition of the improvements we’ve made and your further suggestions. We’ll do our best to incorporate them in the next revision.
>
> Thanks again for your time.
> the authors.

---

### Comment · Area_Chair_YeBH · 2024-11-25
**Please check author response and actively participate in the discussion**

Dear Reviewers,

Thank you for your efforts and contribution to ICLR! The authors have posted their responses to your original comments. Given the limited time remaining for the reviewer-author discussion, your feedback and prompt responses are important. Please actively check the authors' responses and participate in the discussion.

Thanks!

Best regards,

Your AC

---

### Meta-Review · Area_Chair_YeBH · 2024-12-24

**Metareview:**

## Summary:
The paper introduces Contrastive Generative Exploration (CGE) as a method for identifying novel domains within fine-tuning datasets, even when direct access to the data is unavailable. CGE operates by leveraging the disparities in predictions made by a pre-trained model and the same model post-fine-tuning to generate examples that showcase novel properties within the dataset. It generates new data by sampling through a constrained decoding according to the difference of the log probabilities of the two models. To enhance the diversity of generated outputs, an iterative version of CGE is proposed, where previously generated examples are used to update the pre-trained model, leading to a more comprehensive exploration of novel domains.

Through experiments, the effectiveness of CGE is demonstrated in detecting unique characteristics within fine-tuning data, such as identifying toxic languages or new languages. The method proves its resilience even when dealing with fine-tuning processes involving differential privacy techniques. By utilizing the contrastive score, which is the difference in probabilities between the fine-tuned and pre-trained models, CGE successfully uncovers novel content within the dataset, providing a valuable tool for exploring and understanding hidden properties in large, noisy, and confidential datasets used for fine-tuning language models.

## Strengths:
1. The use of the contrastive score to identify new-domain examples in fine-tuning datasets is novel and effective, especially in potentially recovering parts of confidential data that cannot be directly accessed. The new metric for detecting novel content achieves notably high performance, surpassing many existing metrics in this domain.
1. The experimental setup is robust, featuring experiments across various base LLMs and datasets, with comprehensive reporting on multiple evaluation metrics.
1. The iterative version of the CGE framework is well-motivated.

## Weaknesses:
1. More clarifications, justifications, and comparisons to existing approaches are needed regarding data access in the targeted setting.
1. More analysis and comparisons of different proportions of novel data in the finetuning are needed.
1. Some case studies and analysis of the CGE-generated novel data are necessary. It is also important to discuss whether additional quality checks and filtering need to be applied to the sampled data.
1. The writing and clarity of the paper need a lot of improvement, especially the goal and the evaluation.
1. The discussion and experiments regarding DP should be more rigorous.

## Decision:
The authors provided further clarifications and additional experimental results in the rebuttal, as requested by the reviewers. The original review comments raise several critical concerns regarding the clarity of the paper. After the rebuttal, three reviewers out of the four raised their ratings, including one increased from 3 to 6. This reflects the effectiveness of the responses by the authors. The final ratings all vote for accepting this paper (all 6). The meta-reviewer carefully read all the comments, rebuttals, and discussions. Although the meta-reviewer agrees with the reviewers that the revision and responses by the authors greatly improve the original draft, these changes should be carefully organized and included in the draft to avoid the confusion and misunderstanding that the reviewers had before. More discussions on why the proposed method cannot be replaced by other existing approaches are also necessary. Since the paper addressed a novel problem that can be interesting to some specific groups of researchers, and the empirical results do show the advantages over other metrics, the meta-reviewer is glad to recommend the acceptance of this paper.

**Additional Comments On Reviewer Discussion:**

The authors provided further clarifications and additional experimental results in the rebuttal, as requested by the reviewers. The original review comments raise several critical concerns regarding the clarity of the paper. After the rebuttal, three reviewers out of the four raised their ratings, including one increased from 3 to 6. This reflects the effectiveness of the responses by the authors. The final ratings all vote for accepting this paper (all 6).

---

### Decision · Program_Chairs · 2025-01-22

Accept (Poster)